FERMILAB-PUB-22-741-SCD

# New Machine Learning Techniques for Simulation-Based Inference: InferoStatic Nets, Kernel Score Estimation, and Kernel Likelihood Ratio Estimation

Kyoungchul Kong[1], Konstantin T. Matchev[2], Stephen Mrenna[3], and Prasanth Shyamsundar[4*],

[1] Department of Physics and Astronomy, University of Kansas, Lawrence, KS 66045, USA
[2] Institute for Fundamental Theory, Physics Department, University of Florida, Gainesville, FL 32611, USA
[3] Scientific Computing Division, Fermi National Accelerator Laboratory, Batavia, IL 60510, USA
[4] Fermilab Quantum Institute, Fermi National Accelerator Laboratory, Batavia, IL 60510, USA
* prasanth@fnal.gov

October 4, 2022

## Abstract

We propose an intuitive, machine-learning approach to multiparameter inference, dubbed the InferoStatic Networks (ISN) method, to model the score and likelihood ratio estimators in cases when the probability density can be sampled but not computed directly. The ISN uses a backend neural network that models a scalar function called the inferostatic potential $\varphi$. In addition, we introduce new strategies, respectively called Kernel Score Estimation (KSE) and Kernel Likelihood Ratio Estimation (KLRE), to learn the score and the likelihood ratio functions from simulated data. We illustrate the new techniques with some toy examples and compare to existing approaches in the literature. We mention *en passant* some new loss functions that optimally incorporate latent information from simulations into the training procedure.

# 1   Introduction

Inference in physical sciences, such as particle physics, relies on comparing detailed predictions from computationally expensive simulations to data. These predictions depend upon input parameters that are the objects of interest in parameter-estimation analyses. Classical inference techniques for parameter measurement include the analysis of histograms of summary statistics, the matrix element method, optimal observables, etc. (see [1, 2] for recent reviews and a guide to the literature). More recently, there has been an explosion of interest in corresponding Machine Learning (ML) techniques for parameter measurement, which rely only on samples generated at different parameter values. The basic appeal of the ML approach is that it can leverage high-dimensional information not captured by summary statistics. An up-to-date compendium of the literature on ML applications in particle physics is maintained at [3].

The ML problem at hand can be described as follows. Let $\boldsymbol{x} = (x_1, \ldots, x_D)$ be a $D$-dimensional random variable (datapoint; collision event in the context of collider physics) whose unit-normalized distribution under a given theory model is $p(\boldsymbol{x} \,;\, \boldsymbol{\theta})$, where $\boldsymbol{\theta} \equiv (\theta_1, \ldots, \theta_d)$ is a $d$-dimensional continuous parameter of the model. A standard problem, also encountered in high energy physics, is to estimate the value of the parameter $\boldsymbol{\theta}$ using sets of $N$ independent datapoints $\mathcal{X} \equiv \{\boldsymbol{x}_1, \ldots, \boldsymbol{x}_N\}$, with each set produced using the same value of $\boldsymbol{\theta} = \boldsymbol{\theta}_{\text{true}}$.[1] The following definitions are often relevant in the context of such estimations:

$$s(\boldsymbol{x} \,;\, \boldsymbol{\theta}) \equiv \nabla_{\boldsymbol{\theta}} \ln p(\boldsymbol{x} \,;\, \boldsymbol{\theta}), \tag{1a}$$

---

[1]We will assume that the theory model $p$ satisfies all the conditions for the maximum likelihood estimator for $\boldsymbol{\theta}_{\text{true}}$ to be asymptotically consistent, for all possible values of $\boldsymbol{\theta}_{\text{true}}$. Among other things, this ensures that if $p(\boldsymbol{x} \,;\, \boldsymbol{\theta}) = p(\boldsymbol{x} \,;\, \boldsymbol{\theta}')$ almost everywhere, then $\boldsymbol{\theta} = \boldsymbol{\theta}'$.

$$r(\boldsymbol{x} \,;\, \boldsymbol{\theta}_0, \boldsymbol{\theta}_1) \equiv \frac{p(\boldsymbol{x} \,;\, \boldsymbol{\theta}_0)}{p(\boldsymbol{x} \,;\, \boldsymbol{\theta}_1)}, \tag{1b}$$

$$r_{\mathrm{ref}}(\boldsymbol{x} \,;\, \boldsymbol{\theta}) \equiv \frac{p(\boldsymbol{x} \,;\, \boldsymbol{\theta})}{p_{\mathrm{ref}}(\boldsymbol{x})}, \tag{1c}$$

where $\nabla_{\boldsymbol{\theta}}$ represents the $d$-dimensional gradient with respect to $\boldsymbol{\theta}$ and $p_{\mathrm{ref}}$ is a reference distribution. The $d$-dimensional function $\boldsymbol{s}$ is referred to as the score function, $r_{\mathrm{ref}}$ as the "singly parameterized likelihood ratio function" (because it has one $\boldsymbol{\theta}$), and $r$ as the "doubly parameterized likelihood ratio" (because it involves $\boldsymbol{\theta}_0$ and $\boldsymbol{\theta}_1$). In the rest of this paper, the term "likelihood ratio function" refers to the doubly parameterized likelihood ratio $r$, unless otherwise stated.

In many situations, there is no feasible technique to compute $p$ directly, particularly when the dimensionality of $\boldsymbol{x}$ or $\boldsymbol{\theta}$ is large. Nevertheless, there can exist an oracle to produce datapoints distributed according to $p$ for any chosen value of $\boldsymbol{\theta}$. Several approaches have been developed to learn the function $p$ itself using simulated data produced by such an oracle. The learned function $\hat{p}(\boldsymbol{x} \,;\, \boldsymbol{\theta})$ can be used for estimating $\boldsymbol{\theta}_{\mathrm{true}}$ and for a number of other tasks, such as event generation, unfolding [4], and anomaly detection [5]. However, for high-dimensional data, it is often easier to train a neural network to learn the likelihood ratio rather than the likelihood function itself. This motivates alternative approaches which use the simulated data to estimate the functions $\boldsymbol{s}$, $r$, and $r_{\mathrm{ref}}$ over a range of $\boldsymbol{\theta}$ using ML techniques [6–11]. These learned $\boldsymbol{s}$, $r$, and $r_{\mathrm{ref}}$ functions can then be used in the estimation of $\boldsymbol{\theta}_{\mathrm{true}}$ from experimental data, as well as for other related tasks, as reviewed in Section 1.1, using standard methods like gradient descent, *etc.* In this paper, we introduce some new ML strategies to learn the score function $\boldsymbol{s}$ and likelihood ratio function $r$ from the simulated data, particularly in those situations where the function $p$ can be sampled but not computed directly.

## 1.1 Applications of Estimated Scores and Likelihood Ratios

Here we briefly review some applications of the score function $\boldsymbol{s}$ and likelihood ratio functions $r_{\mathrm{ref}}$ and $r$ after they are estimated from simulations.

**Direct parameter estimation.** The unknown value of $\boldsymbol{\theta}$ can be estimated from an experimental dataset $\{\boldsymbol{x}_i\}_{i=1}^{N}$ of $N$ points, sampled independently from $p(\boldsymbol{x} \,;\, \boldsymbol{\theta}_{\mathrm{true}})$, using the parameterized likelihood ratio functions and/or the score function. For example, the maximum likelihood estimator can be written as [6]

$$\hat{\boldsymbol{\theta}}_{\mathrm{MLE}} = \arg\min_{\boldsymbol{\theta}'} \left[ -\frac{1}{N} \sum_{i=1}^{N} \ln r_{\mathrm{ref}}(\boldsymbol{x}_i \,;\, \boldsymbol{\theta}') \right]. \tag{2}$$

Similarly, Ref. [12] showed how to estimate $\boldsymbol{\theta}$ using the binary cross-entropy loss function, written as

$$\hat{\boldsymbol{\theta}}_{\mathrm{BCE}} = \arg\min_{\boldsymbol{\theta}'} \left[ -\frac{1}{N} \sum_{i=1}^{N} \ln \left( \frac{r_{\mathrm{ref}}(\boldsymbol{x}_i \,;\, \boldsymbol{\theta}')}{1 + r_{\mathrm{ref}}(\boldsymbol{x}_i \,;\, \boldsymbol{\theta}')} \right) - \frac{1}{N_{\mathrm{ref}}} \sum_{i=1}^{N_{\mathrm{ref}}} \ln \left( \frac{1}{1 + r_{\mathrm{ref}}(\boldsymbol{x}_i^{\mathrm{ref}} \,;\, \boldsymbol{\theta}')} \right) \right], \tag{3}$$

which uses the experimental data $\{\boldsymbol{x}_i\}_{i=1}^{N}$ produced under the true unknown $\boldsymbol{\theta}_{\mathrm{true}}$ *and* the additional $\{\boldsymbol{x}_i^{\mathrm{ref}}\}_{i=1}^{N_{\mathrm{ref}}}$ simulated dataset produced under a reference value $\boldsymbol{\theta}_{\mathrm{ref}}$.

Eqs. (2) and (3) provide two ways for using the $r_{\mathrm{ref}}$ function to estimate $\boldsymbol{\theta}$. Furthermore, if the optimization in (2) and (3) is to be performed using gradient-based techniques, the score

function implicitly becomes relevant. The gradient of the objective function in (2) with respect to $\boldsymbol{\theta}'$ is given by

$$\nabla_{\boldsymbol{\theta}'}\left[-\frac{1}{N}\sum_{i=1}^{N}\ln r_{\text{ref}}(\boldsymbol{x}_i\,;\,\boldsymbol{\theta}')\right]=-\frac{1}{N}\sum_{i=1}^{N}\boldsymbol{s}(\boldsymbol{x}_i\,;\,\boldsymbol{\theta}').\tag{4}$$

Likewise, the gradients of the two terms in (3) are given by

$$\nabla_{\boldsymbol{\theta}'}\left[-\ln\left(\frac{r_{\text{ref}}(\boldsymbol{x}\,;\,\boldsymbol{\theta}')}{1+r_{\text{ref}}(\boldsymbol{x}\,;\,\boldsymbol{\theta}')}\right)\right]=\frac{-1}{1+r_{\text{ref}}(\boldsymbol{x}\,;\,\boldsymbol{\theta}')}\,\boldsymbol{s}(\boldsymbol{x}\,;\,\boldsymbol{\theta}'),\tag{5a}$$

$$\nabla_{\boldsymbol{\theta}'}\left[-\ln\left(\frac{1}{1+r_{\text{ref}}(\boldsymbol{x}\,;\,\boldsymbol{\theta}')}\right)\right]=\frac{r_{\text{ref}}(\boldsymbol{x}\,;\,\boldsymbol{\theta}')}{1+r_{\text{ref}}(\boldsymbol{x}\,;\,\boldsymbol{\theta}')}\,\boldsymbol{s}(\boldsymbol{x}\,;\,\boldsymbol{\theta}').\tag{5b}$$

Equations (2)-(5) show how the score function or the singly parameterized likelihood ratio function could be used to perform the maximum likelihood estimation of $\boldsymbol{\theta}$.

In the context of high energy physics, this technique can be used to estimate either theory parameters or nuisance parameters. The former is usually referred to as parameter measurement, while the latter is referred to as parameter tuning [13]. Theory parameter measurement and nuisance parameter tuning often have different requirements and standards on a) uncertainty quantification, b) interpretability of the estimation technique, and c) how validatable the simulation models are for the purposes of the chosen estimation technique. For example one should opt for highly validatable estimation techniques for theory parameter measurements, in order to be robust against unknown errors in the simulation-models (i.e., not accounted for by known systematic uncertainties). On the other hand, nuisance parameter tuning methods should ensure that the systematic uncertainties corresponding to the relevant nuisance parameters are not underestimated in final results.

**Locally optimal observables.** The score function evaluated at $\boldsymbol{\theta}=\boldsymbol{\theta}_0$ is a sufficient statistic, i.e., optimal variable, for the estimation of a parameter $\boldsymbol{\theta}$ near $\boldsymbol{\theta}_0$ [14]. In this way, the learned score $\hat{\boldsymbol{s}}$ can be used as an optimal analysis variable, provided that one expects the true value to be in the vicinity of $\boldsymbol{\theta}_0$.

**Dataset reweighting.** The knowledge of the parameterized likelihood ratio functions allow us to reweight events produced under one value of $\boldsymbol{\theta}$, say $\boldsymbol{\theta}_0$, to emulate a dataset produced under a different value, say $\boldsymbol{\theta}_1$ [15,16]. In this case, the appropriate weighting function will be

$$\text{weight}(\boldsymbol{x})=\frac{p(\boldsymbol{x}\,;\,\boldsymbol{\theta}_1)}{p(\boldsymbol{x}\,;\,\boldsymbol{\theta}_0)}=r(\boldsymbol{x}\,;\,\boldsymbol{\theta}_1,\boldsymbol{\theta}_0).\tag{6}$$

## 1.2 Related Techniques and New Contributions in This Work

This work builds on a previous, related body of knowledge [1, 6–12]. To provide some context, Table 1 lists some of the existing simulation-based score, likelihood, and likelihood ratio estimators, categorizing them according to i) which of the three quantities in (1) they estimate (rows) and ii) whether or not they use additional latent information from the simulators (columns). Four new and distinct contributions are presented in this work, listed below.

1. In Section 2, we propose an intuitive approach to *model* the estimators $\hat{\boldsymbol{s}}(\boldsymbol{x}\,;\,\boldsymbol{\theta})$ and $\hat{r}(\boldsymbol{x}\,;\,\boldsymbol{\theta}_0,\boldsymbol{\theta}_1)$ via a backend neural network for a scalar function $\hat{\varphi}(\boldsymbol{x},\boldsymbol{\theta})$. This approach, dubbed the **InferoStatic Networks method** (ISN), offers some advantages over directly modeling $\hat{\boldsymbol{s}}$ and $\hat{r}$ using neural networks.

Table 1: The landscape of the simulation-based score and likelihood ratio estimators described in this paper and the existing approaches in the literature. The ISN approach described in Section 2 can be applied to all these cases.

| | Only requires observable data from the simulator | Requires additional latent simulation information |
|---|---|---|
| Singly parameterized likelihood estimator | • NDE [7]<br>• MEM [17] | • MadMiner [8] [SCANDAL] |
| Singly parameterized likelihood ratio estimator (to a reference distribution) | • MadMiner [6] [CARL]<br>• DCTR [12] | • MadMiner [ROLR, ALICE, CASCAL, RASCAL, ALICES] |
| Doubly parameterized likelihood ratio estimation | • MadMiner [CARL]<br>• KLRE [Section 4] | • MadMiner [ROLR, ALICE, CASCAL, RASCAL, ALICES]<br>• This work [Appendix A] |
| Score estimator | • KSE [Section 3] | • MadMiner [9]<br>[SALLY, SALLINO] |

2. In Section 3, we introduce a technique, dubbed **Kernel Score Estimation** (KSE), to *train* a network to learn the score function $s$ from simulated data.

3. In Section 4, we introduce a technique, dubbed **Kernel Likelihood Ratio Estimation** (KLRE), to learn the doubly parameterized likelihood ratio function $r$ from simulated data. This technique generalizes the previously known CARL technique for learning $r$ [6].

4. In Appendix A, we provide some new loss functions for incorporating additional latent information from the simulation pipeline into the training of $\hat{r}$.

In Section 5, we illustrate the new techniques with some toy examples and compare to the corresponding approaches already existing in the literature. Section 6 is reserved for our conclusions. Several technical discussions and derivations are collected in the appendices.

## 2 Methodology: InferoStatic Networks (ISNs)

This work is focused on developing ML techniques to infer the score or likelihood ratio. The standard approach in the literature is to model $\hat{s}(x\,;\,\theta)$ and $\hat{r}(x\,;\,\theta_0,\theta_1)$ directly as neural networks. However, inspired by the definitions of $s$ and $r$ in (1), we propose to use a neural network to model a scalar function $\hat{\varphi}(x,\theta)$, and define $\hat{s}$ and $\hat{r}$ via $\hat{\varphi}$ as

$$\hat{s}(x\,;\,\theta) \equiv \nabla_\theta\,\hat{\varphi}(x,\theta), \tag{7a}$$

$$\hat{r}(x\,;\,\theta_0,\theta_1) \equiv \exp\left[\hat{\varphi}(x,\theta_0) - \hat{\varphi}(x,\theta_1)\right] = \frac{\exp\left[\hat{\varphi}(x,\theta_0)\right]}{\exp\left[\hat{\varphi}(x,\theta_1)\right]}. \tag{7b}$$

Here $\hat{\varphi}$ plays the same role in the definitions of $\hat{s}$ and $\hat{r}$ as $\ln p$ does in the definitions of $s$ and $r$ in (1). We dub $\hat{\varphi}$ as the "inferostatic potential", in analogy with the electrostatic potential from

physics.[2] Similarly, the neural networks for $\hat{\varphi}$, $\hat{s}$, and $\hat{r}$ will be referred to collectively as Infero-Static Networks (ISNs). Individually, they will be referred to as inferostatic potential, inferostatic score, and inferostatic likelihood-ratio networks, respectively (see Figure 1). As a simple example, we can model $\hat{\varphi}$ as a single artificial neuron with a linear activation:

$$\hat{\varphi}(\boldsymbol{x}, \boldsymbol{\theta}) \equiv \boldsymbol{u} \cdot \boldsymbol{x} + \boldsymbol{v} \cdot \boldsymbol{\theta}, \tag{8}$$

where $\boldsymbol{u}$ and $\boldsymbol{v}$ are $D$- and $d$-dimensional tunable parameters of the neuron, respectively. In this case, $\hat{s}$ and $\hat{r}$ will be given by

$$\hat{s}(\boldsymbol{x}\,;\,\boldsymbol{\theta}) = \boldsymbol{v}, \qquad\qquad \hat{r}(\boldsymbol{x}\,;\,\boldsymbol{\theta}, \boldsymbol{\theta}') = \exp\left[\boldsymbol{v} \cdot \left(\boldsymbol{\theta} - \boldsymbol{\theta}'\right)\right]. \tag{9}$$

From (1) and (7), it can be seen that the following three situations or conditions are equivalent:

1. The neural network function $\hat{\varphi}$ matches $\ln p + c$, where $c$ is an arbitrary function of $\boldsymbol{x}$ only (i.e., independent of $\boldsymbol{\theta}$).

2. The estimated score function $\hat{s}$ matches the true score function $\boldsymbol{s}$.

3. The estimated likelihood-ratio function $\hat{r}$ matches the true likelihood-ratio $r$.

An important property of inferostatic score networks is that if $\hat{\varphi}(\boldsymbol{x}, \boldsymbol{\theta})$ is modeled as a feed-forward neural network with trainable weights $\boldsymbol{w}$, then its gradient $\hat{s}(\boldsymbol{x}\,;\,\boldsymbol{\theta})$ can also be expressed as a feed-forward network with the same trainable weights $\boldsymbol{w}$, as shown in Appendix B. This allows backpropagation to be used for training the weights of the score network. The construction of the score network for $\hat{s}$ from the potential network for $\hat{\varphi}$ can be automated under modern machine learning platforms that support auto-differentiation, e.g., `TensorFlow` [18].

Note that the ISN approach of modeling $\hat{s}$ and $\hat{r}$ is compatible not just with the training techniques introduced in the subsequent sections of this paper, but also with other techniques in the literature for training scores and doubly parameterized likelihood ratios, including all the relevant techniques implemented in MadMiner [11]. The ISN approach offers several advantages as discussed below.

**Building properties of $s$ and $r$ into their estimators**

By modeling $\hat{s}$ as a gradient and $\hat{r}$ as a ratio, certain properties of the score function $\boldsymbol{s}$ and likelihood ratio $r$ are built into their corresponding estimators under our approach:

$$\oint_C \hat{s}(\boldsymbol{x}\,;\,\boldsymbol{\theta}) \cdot d\boldsymbol{\theta} = 0, \qquad \forall\forall \boldsymbol{x}, \text{ along any closed path } C \text{ in the } \boldsymbol{\theta} \text{ space}, \tag{10a}$$

$$\hat{r}(\boldsymbol{x}\,;\,\boldsymbol{\theta}_0, \boldsymbol{\theta}_1) \geq 0, \qquad\qquad\qquad \forall \boldsymbol{x}, \boldsymbol{\theta}_0, \boldsymbol{\theta}_1, \tag{10b}$$

$$\hat{r}(\boldsymbol{x}\,;\,\boldsymbol{\theta}_0, \boldsymbol{\theta}_1)\,\hat{r}(\boldsymbol{x}\,;\,\boldsymbol{\theta}_1, \boldsymbol{\theta}_2) = \hat{r}(\boldsymbol{x}\,;\,\boldsymbol{\theta}_0, \boldsymbol{\theta}_2), \qquad \forall \boldsymbol{x}, \boldsymbol{\theta}_0, \boldsymbol{\theta}_1, \boldsymbol{\theta}_2, \tag{10c}$$

$$\hat{r}(\boldsymbol{x}\,;\,\boldsymbol{\theta}_0, \boldsymbol{\theta}_1) = \left[\hat{r}(\boldsymbol{x}\,;\,\boldsymbol{\theta}_1, \boldsymbol{\theta}_0)\right]^{-1}, \qquad \forall \boldsymbol{x}, \boldsymbol{\theta}_0, \boldsymbol{\theta}_1, \tag{10d}$$

$$\hat{r}(\boldsymbol{x}\,;\,\boldsymbol{\theta}, \boldsymbol{\theta}) = 1, \qquad\qquad\qquad \forall \boldsymbol{x}, \boldsymbol{\theta}, \tag{10e}$$

---

[2]Up to an overall sign, the electrostatic potential relates changes in the electric potential to the electric field in the same way as the inferostatic potential $\hat{\varphi}$ relates the log-likelihood ratio estimate $\ln \hat{r}$ to the score estimate $\hat{s}$ in (7). While electric fields and potentials are concerned with electric charge, $\hat{\varphi}$, $\hat{s}$, and $\hat{r}$ are concerned with parameter inference, hence the name "inferostatic".

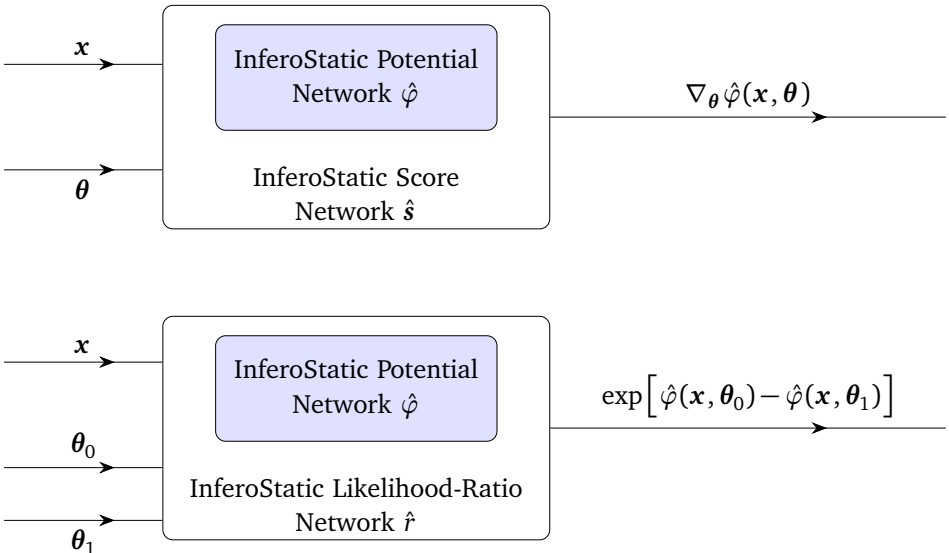

Figure 1: Schematic diagrams representing the inferostatic score network (upper diagram) and the inferostatic likelihood-ratio network (lower diagram).

$$\nabla_{\boldsymbol{\theta}_0} \ln \hat{r}(\boldsymbol{x} ; \boldsymbol{\theta}_0, \boldsymbol{\theta}_1) = \hat{s}(\boldsymbol{x} ; \boldsymbol{\theta}_0), \qquad\qquad \forall\forall \boldsymbol{x}, \boldsymbol{\theta}_0, \boldsymbol{\theta}_1 , \tag{10f}$$

where $\forall\forall$ means "for almost all". Directly modeling $\hat{s}$ and $\hat{r}$ as (separate) neural networks, as commonly done in the literature, does not guarantee that all these relations will be satisfied exactly, even after training the corresponding networks. In addition to the associated conceptual elegance, exactly satisfying (10) using our approach offers some technical advantages as well.

1. For example, property (10c) allows for the ISN to extrapolate a good approximation for $r(\boldsymbol{x} ; \boldsymbol{\theta}_0, \boldsymbol{\theta}_2)$ from $\hat{r}(\boldsymbol{x} ; \boldsymbol{\theta}_0, \boldsymbol{\theta}_1)$ and $\hat{r}(\boldsymbol{x} ; \boldsymbol{\theta}_1, \boldsymbol{\theta}_2)$. Each training datapoint with input-value $(\boldsymbol{x}, \boldsymbol{\theta}_0, \boldsymbol{\theta}_1)$ provides information not just on the value of $r$ for that input, but also on the value of $r$ at other inputs of the form $(\boldsymbol{x}, \boldsymbol{\theta}_0, \boldsymbol{\theta}')$ or $(\boldsymbol{x}, \boldsymbol{\theta}', \boldsymbol{\theta}_1)$. ISNs can use the available information more efficiently, which can potentially lead to a more efficient training of the NN-based function $\hat{r}$, in comparison to standard NN modeling approaches that do not enforce (10c).

   Furthermore, by enforcing property (10c), we are facilitating the generalizability of the NN-based function $\hat{r}$ in regions of the input-space that are not well represented in the training dataset, but can nevertheless be extrapolated from the training data. This is illustrated in our example in Section 5.

2. For values of $\boldsymbol{\theta}_1$ sufficiently close to $\boldsymbol{\theta}_0$, the value of $r(\boldsymbol{x} ; \boldsymbol{\theta}_0, \boldsymbol{\theta}_1)$ will be approximately 1. This property is built into the ISN for $\hat{r}$, as exemplified by property (10e), and does not have to be learned by the network from data. This way, by suppressing some of the noise in the function $\hat{r}$, our approach could lead to a more efficient and accurate training of $\hat{r}$, especially for neighboring values of $\boldsymbol{\theta}_0$ and $\boldsymbol{\theta}_1$. Improving the estimation of the function $r$ for neighboring parameter values will lead to a) better resolution (or error) in the subsequent parameter measurement using $\hat{r}$, and b) more accurate reweighting of datapoints between neighboring parameter values.

The two points above only describe some ways in which our structured approach to modeling $\hat{s}$ and $\hat{r}$ via $\hat{\varphi}$ *could* improve the training efficiency. In general, the efficiency of neural-network-training depends on several factors, including the neural network architecture, computational tools and framework, the specific metric used to quantify training efficiency, the specific usage example or application under consideration, *etc*.

**Portability and Complementary Training**

The techniques to train $\hat{s}$ and $\hat{r}$ to match $s$ and $r$, respectively, can be viewed as different techniques to train the common backend function $\hat{\varphi}(\boldsymbol{x}, \boldsymbol{\theta})$ to match $\ln p(\boldsymbol{x} ; \boldsymbol{\theta})$ up to an additive factor of $c(\boldsymbol{x})$. This way, one can port a network trained using the score-learning-techniques to extract likelihood ratios, and vice-versa, at least in principle. This is illustrated in our example in Section 5. Furthermore, the different training techniques from Table 1 can be used in a complementary fashion to train $\hat{\varphi}$. This idea of complementary training was used previously in MadMiner (in the CASCAL, RASCAL, ALICES, and SCANDAL methods), even if a) it was not presented in terms of a common, portable, backend-network $\hat{\varphi}$, and b) it was only used in cases where additional latent information was available.

To summarise this section, the advantages of training the inferostatic scalar are that it automatically enforces constraints that are only approximately satisfied in other methods and that it allows a straightforward application to other tasks.

## 3   Methodology: Kernel Score Estimation

The Kernel Score Estimation (KSE) technique introduced in this section is a new way to estimate the score $\boldsymbol{s}$, and is compatible with any existing architectures for modeling $\hat{s}$, including the ISN architecture introduced in Section 2. Previously, [9] showed how to estimate $\boldsymbol{s}$, but only for cases when additional latent information is available.

### 3.1   Intuition and Motivation

KSE can be thought of as a Monte Carlo-based numerical differentiation of $\ln p(\boldsymbol{x} ; \boldsymbol{\theta})$ with respect to $\boldsymbol{\theta}$. The score (which is the gradient of the objective function) at a given value of the parameter, say $\boldsymbol{\theta}_0$, is assumed to be approximately constant in a sufficiently small neighborhood around $\boldsymbol{\theta}_0$. Our technique involves estimating, from simulated data, the variation in the value of $p(\boldsymbol{x} ; \boldsymbol{\theta})$, for a given $\boldsymbol{x}$, when $\boldsymbol{\theta}$ is sampled from the neighborhood of $\boldsymbol{\theta}_0$. The score $\boldsymbol{s}$ at $\boldsymbol{\theta}_0$ can subsequently be extracted from this variation in $p(\boldsymbol{x} ; \boldsymbol{\theta})$.

This intuition can be strengthened with the following concrete example for a 1-dimensional parameter $\theta$, i.e., $d = 1$. We are interested in estimating the score for a given value of $(\theta, \boldsymbol{x})$, say $(\theta_0, \boldsymbol{x}_0)$. Since the score measures how $p(\boldsymbol{x} ; \theta)$ varies with $\theta$, we sample $\theta'$ uniformly in the range $[\theta_0 - \lambda, \theta_0 + \lambda)$, as shown in the top panel of Figure 2. We then set $\theta = \theta'$ in the simulator

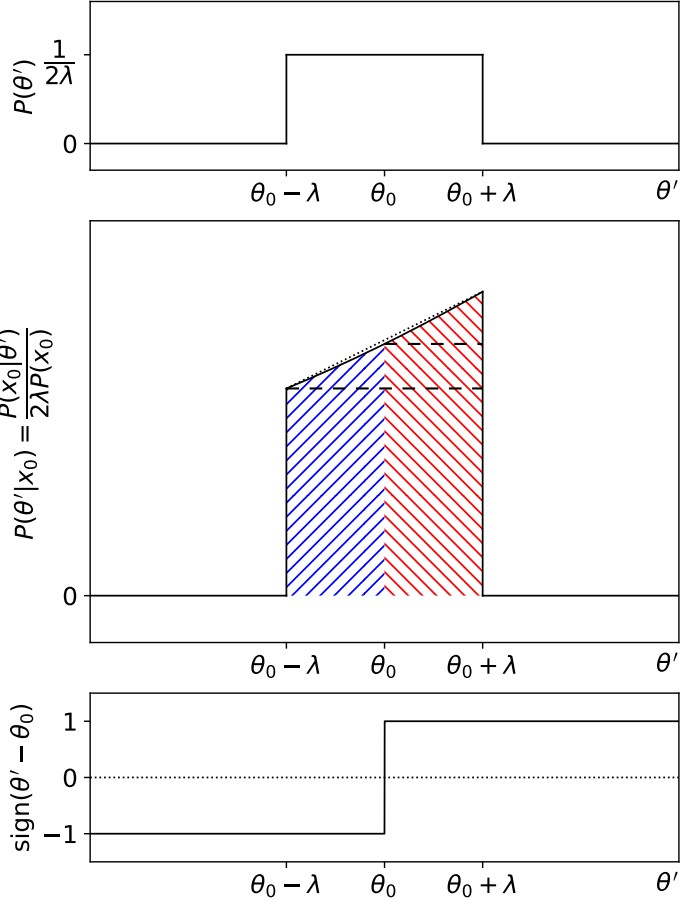

Figure 2: Illustration of the sampling procedure leading up to the score approximation in (14).

and sample event $\boldsymbol{x}$ from $p(\boldsymbol{x}\,;\,\theta')$. The joint probability of $(\theta',\boldsymbol{x})$ is given by

$$
P(\theta',\boldsymbol{x}) = \begin{cases} \dfrac{1}{2\lambda}\, p(\boldsymbol{x}\,;\,\theta'), & \text{if } \theta_0 - \lambda \leq \theta' < \theta_0 + \lambda\,, \\ 0\,, & \text{otherwise}\,. \end{cases} \tag{11}
$$

From this, we can write the probability of $\theta'$ given $\boldsymbol{x} = \boldsymbol{x}_0$ under this data sampling scheme as

$$
P(\theta'\,|\,\boldsymbol{x}_0) = \frac{P(\theta',\boldsymbol{x}_0)}{P(\boldsymbol{x}_0)} = \begin{cases} \dfrac{1}{2\lambda\,P(\boldsymbol{x}_0)}\, p(\boldsymbol{x}_0\,;\,\theta'), & \text{if } \theta_0 - \lambda \leq \theta' < \theta_0 + \lambda\,, \\ 0\,, & \text{otherwise}\,. \end{cases} \tag{12}
$$

A cartoon of this distribution is illustrated as a solid black-curve in the middle panel of Figure 2. If $\lambda$ is chosen sufficiently small, $P(\theta'\,|\,\boldsymbol{x}_0)$ will be approximately linear between $\theta_0 - \lambda$ and $\theta_0 + \lambda$, as indicated in the plot (the hard-to-see slanted dotted black line). The slope of this distribution is directly related to the score function $\partial_\theta \ln p(\boldsymbol{x}_0\,;\,\theta)\big|_{\theta=\theta_0}$. Now consider the difference $\Delta A$ between a) the area under the curve from $\theta_0$ to $\theta_0 + \lambda$ (red backslash hatches), and b) the area under the curve from $\theta_0 - \lambda$ to $\theta_0$ (blue forwardslash hatches). A positive (negative) difference $\Delta A$

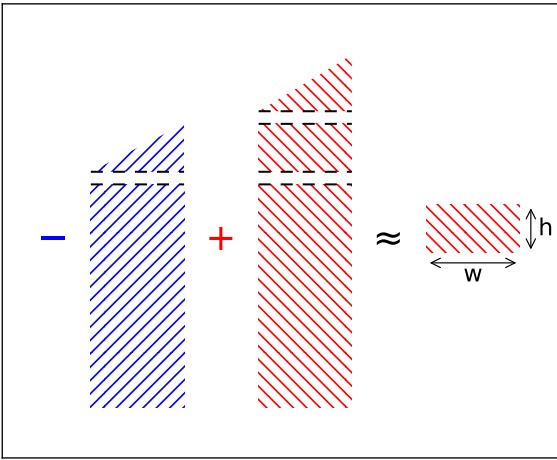

Figure 3: A pictorial illustration of the shape math in (13).

is indicative of a positive (negative) score at $(\boldsymbol{x}_0, \theta_0)$. Note that $\Delta A$ can be expressed as the expectation under $P(\theta' | \boldsymbol{x}_0)$ of the sign function depicted in the bottom panel of Figure 2. With this setup, the area difference $\Delta A$, derived pictorially in Figure 3, can be written as

$$\Delta A = \mathrm{E}_P \left[ \mathrm{sign}(\theta' - \theta_0) \, \Big| \, \boldsymbol{x}_0 \right] \approx \mathrm{w} \times \mathrm{h} \text{ (in the right hand side of Figure 3)} \tag{13a}$$

$$\approx \lambda \times \frac{\lambda \, \partial_\theta p(\boldsymbol{x}_0\,;\,\theta)\Big|_{\theta=\theta_0}}{2\lambda \, P(\boldsymbol{x}_0)} \approx \frac{\lambda}{2} \, \frac{\partial_\theta p(\boldsymbol{x}_0\,;\,\theta)\Big|_{\theta=\theta_0}}{p(\boldsymbol{x}_0\,;\,\theta_0)} = \frac{\lambda}{2} \, s(\boldsymbol{x}_0\,;\,\theta_0). \tag{13b}$$

This lets us rewrite the score as

$$s(\boldsymbol{x}\,;\,\theta_0) \approx \frac{\mathrm{E}_P \left[ \mathrm{sign}(\theta' - \theta_0) \, \Big| \, \boldsymbol{x} \right]}{\lambda/2} = \frac{\mathrm{E}_P \left[ \mathrm{sign}(\theta' - \theta_0) \, \Big| \, \boldsymbol{x} \right]}{\mathrm{E}_P \left[ (\theta' - \theta) \, \mathrm{sign}(\theta' - \theta_0) \right]}. \tag{14}$$

All the approximations in the previous two equations are exact in the limit $\lambda \to 0$. Note that the denominator in (14) is independent of the probability distribution $p$, and can be calculated based on the sampling scheme for producing $\theta'$. The numerator can be estimated using regression techniques, with $\boldsymbol{x}$ being the input and $\mathrm{sign}(\theta' - \theta_0)$ being the regression target.

This intuition forms the basis of the Kernel Score Estimation (KSE) technique, which incorporates the following generalizations:

- The rectangular kernel can be replaced with a different symmetric kernel distribution to sample $\theta'$ around $\theta_0$.

- Similarly, $\mathrm{sign}(\theta' - \theta_0)$ can be replaced with another odd, "difference" function $\psi(\theta' - \theta_0)$.

- The score can be estimated for multi-dimensional parameters $\boldsymbol{\theta}$.

- The score can be estimated at multiple values of the parameter $\boldsymbol{\theta}$ (instead of only at $\boldsymbol{\theta}_0$) using training data produced for different $\boldsymbol{\theta}$ values.

In Section 3.2, we provide the Kernel Score Approximation (KSA), which is the generalized form of (14), before describing how to use KSA with machine learning to estimate scores in Section 3.3.

Note, another possible method for estimating the score function is to first estimate the singly parameterized likelihood ratio function as $\hat{r}_{\text{ref}}$, e.g., using the CARL technique, and then take the gradient of $\ln \hat{r}_{\text{ref}}(x\,;\boldsymbol{\theta})$ with respect to $\boldsymbol{\theta}$ [12] as the score estimate. However, if the estimate $\ln \hat{r}_{\text{ref}}$ for the true log-likelihood ratio function $\ln r_{\text{ref}}$ has a small, but rapidly-changing error, $\nabla_{\boldsymbol{\theta}} \ln \hat{r}_{\text{ref}}(x\,;\boldsymbol{\theta})$ will be a poor estimate for the score $s(x\,;\boldsymbol{\theta}) = \nabla_{\boldsymbol{\theta}} \ln r_{\text{ref}}(x\,;\boldsymbol{\theta})$, even if $\hat{r}_{\text{ref}}$ is a good estimate for $r_{\text{ref}}$, according to the metrics used to evaluate and train $\hat{r}_{\text{ref}}$.[3] Such rapidly changing errors in $\ln \hat{r}_{\text{ref}}$ can be suppressed by incorporating sufficient regularization in the training of $\hat{r}_{\text{ref}}$ (which, in turn, can introduce a bias in the estimated $\hat{r}_{\text{ref}}$). On the other hand, our KSE technique offers a way to avoid this problem entirely, by directly training an estimate $\hat{s}$ for the score $s$.

## 3.2 Kernel Score Approximation

**Setup:** Let $\boldsymbol{\epsilon} = (\epsilon_1, \epsilon_2, \ldots, \epsilon_d)$ be a $d$-dimensional parameter used to denote displacements from the parameter $\boldsymbol{\theta}$ (also $d$-dimensional). Let $K_{\boldsymbol{\theta}}(\boldsymbol{\epsilon})$ be a unit-normalized probability distribution for $\boldsymbol{\epsilon}$ that is i) symmetric around $\mathbf{0}$ in each of the $d$ directions, and ii) possibly parameterized by $\boldsymbol{\theta}$. More concretely, consider a reflection, denoted as $\text{Ref}_j$, with respect to a hyperplane in $\mathbb{R}^d$ which passes through the origin and is orthogonal to the unit vector $\hat{\mathbf{e}}_j$ corresponding to the $j$-th axis. For any $j = 1, 2, \ldots, d$, the reflection $\text{Ref}_j$ transforms the $d$-dimensional vector $\boldsymbol{\epsilon}$ as

$$\text{Ref}_j(\boldsymbol{\epsilon}) \equiv \boldsymbol{\epsilon} - 2(\boldsymbol{\epsilon} \cdot \hat{\mathbf{e}}_j)\hat{\mathbf{e}}_j = \left(\epsilon_1, \ldots, -\epsilon_j, \ldots, \epsilon_d\right). \tag{15}$$

The kernel $K$ is invariant under this reflection operation:

$$K_{\boldsymbol{\theta}}\left(\text{Ref}_j(\boldsymbol{\epsilon})\right) = K_{\boldsymbol{\theta}}\left(\boldsymbol{\epsilon}\right), \quad \forall \boldsymbol{\theta}, \boldsymbol{\epsilon} \in \mathbb{R}^d, j \in \{1, \ldots, d\}. \tag{16}$$

A simple example of such a symmetric kernel in $d$ dimensions is given by

$$K_{\boldsymbol{\theta}}^{\text{rect}}(\boldsymbol{\epsilon}) = \prod_{i=1}^{d} \frac{1}{\lambda_i(\boldsymbol{\theta})} \, \text{rect}\left(\frac{\epsilon_i}{\lambda_i(\boldsymbol{\theta})}\right), \tag{17}$$

where rect is the rectangular function

$$\text{rect}(u) \equiv \begin{cases} 0.5, & \text{if } -1 \leq u < 1, \\ 0, & \text{otherwise}, \end{cases} \tag{18}$$

and the $\lambda_i(\boldsymbol{\theta})$ are positive width parameters. Another possible choice is the delta kernel

$$K_{\boldsymbol{\theta}}^{\text{delta}}(\boldsymbol{\epsilon}) = \frac{1}{2^d} \prod_{i=1}^{d} \delta_{\text{Dirac}}(|\epsilon_i| - \lambda_i(\boldsymbol{\theta})), \tag{19}$$

where $\delta_{\text{Dirac}}$ is the Dirac delta function.

---

[3]This is reflected in the fact that even the uniform convergence of a sequence of differentiable functions, say $(\hat{f}_1, \hat{f}_2, \ldots)$, to a differentiable limiting function, say $f$, does not imply that the derivatives of $(\hat{f}_1, \hat{f}_2, \ldots)$ will converge, uniformly or pointwise, to the derivative of $f$.

Let $\boldsymbol{\psi}_{\boldsymbol{x},\boldsymbol{\theta}} = (\psi_{\boldsymbol{x},\boldsymbol{\theta},1},\dots,\psi_{\boldsymbol{x},\boldsymbol{\theta},i},\dots,\psi_{\boldsymbol{x},\boldsymbol{\theta},d})$ be a $d$-dimensional function of $\boldsymbol{\epsilon}$, possibly parame-terized by $\boldsymbol{x}$ and $\boldsymbol{\theta}$, which transforms under $\mathrm{Ref}_j$ as

$$\psi_{\boldsymbol{x},\boldsymbol{\theta},i}\big(\mathrm{Ref}_j(\boldsymbol{\epsilon})\big) = \begin{cases} -\psi_{\boldsymbol{x},\boldsymbol{\theta},i}(\boldsymbol{\epsilon}), & \text{if } i = j, \\ +\psi_{\boldsymbol{x},\boldsymbol{\theta},i}(\boldsymbol{\epsilon}), & \text{if } i \neq j, \end{cases} \qquad \forall \boldsymbol{\theta}, \boldsymbol{\epsilon} \in \mathbb{R}^d. \tag{20}$$

In other words, the $i$-th component of $\boldsymbol{\psi}_{\boldsymbol{x},\boldsymbol{\theta}}$ is an odd function of $\epsilon_i$, and an even function of the other components of $\boldsymbol{\epsilon}$.

$$\psi_{\boldsymbol{x},\boldsymbol{\theta},i}(\boldsymbol{\epsilon}) \mapsto -\psi_{\boldsymbol{x},\boldsymbol{\theta},i}(\boldsymbol{\epsilon}) \ \text{ under } \ \epsilon_i \mapsto -\epsilon_i, \qquad \forall i, \forall \boldsymbol{x}, \forall \boldsymbol{\theta}, \forall \boldsymbol{\epsilon}, \tag{21a}$$

$$\psi_{\boldsymbol{x},\boldsymbol{\theta},i}(\boldsymbol{\epsilon}) \mapsto +\psi_{\boldsymbol{x},\boldsymbol{\theta},i}(\boldsymbol{\epsilon}) \ \text{ under } \ \epsilon_j \mapsto -\epsilon_j, \qquad \forall j \neq i, \forall \boldsymbol{x}, \forall \boldsymbol{\theta}, \forall \boldsymbol{\epsilon}. \tag{21b}$$

A simple example is the linear function

$$\psi_{\boldsymbol{x},\boldsymbol{\theta},i}^{(\text{linear})}(\boldsymbol{\epsilon}) \equiv \epsilon_i. \tag{22}$$

**KSA Formula:** Let $\mathcal{P}$ be the unit-normalized joint-distribution of the triplet $(\boldsymbol{x},\boldsymbol{\theta},\boldsymbol{\epsilon})$ given by

$$\mathcal{P}(\boldsymbol{x},\boldsymbol{\theta},\boldsymbol{\epsilon}) \equiv \pi(\boldsymbol{\theta}) \cdot K_{\boldsymbol{\theta}}(\boldsymbol{\epsilon}) \cdot p(\boldsymbol{x}\,;\,\boldsymbol{\theta}+\boldsymbol{\epsilon}), \tag{23}$$

where $\pi(\boldsymbol{\theta})$ is a unit-normalized prior for $\boldsymbol{\theta}$. Based on the discussion in Section 3.1 (see eq. (14)), here we introduce the following approximation for the score function $\boldsymbol{s}$, dubbed the Kernel Score Approximation $\boldsymbol{s}^{\text{KSA}}$:

$$s_i(\boldsymbol{x}\,;\,\boldsymbol{\theta}) \approx s_i^{\text{KSA}}(\boldsymbol{x}\,;\,\boldsymbol{\theta}) \equiv \frac{\mathrm{E}_{(\boldsymbol{x},\boldsymbol{\theta},\boldsymbol{\epsilon})\sim\mathcal{P}}\big[\psi_{\boldsymbol{x},\boldsymbol{\theta},i}(\boldsymbol{\epsilon}) \,\big|\, \boldsymbol{x},\boldsymbol{\theta}\big]}{\mathrm{E}_{\boldsymbol{\epsilon}\sim K_{\boldsymbol{\theta}}}\big[\epsilon_i\,\psi_{\boldsymbol{x},\boldsymbol{\theta},i}(\boldsymbol{\epsilon})\big]}, \tag{24a}$$

$$= \mathrm{E}_{(\boldsymbol{x},\boldsymbol{\theta},\boldsymbol{\epsilon})\sim\mathcal{P}}\left[\frac{\psi_{\boldsymbol{x},\boldsymbol{\theta},i}(\boldsymbol{\epsilon})}{\mathrm{E}_{\boldsymbol{\epsilon}\sim K_{\boldsymbol{\theta}}}\big[\epsilon_i\,\psi_{\boldsymbol{x},\boldsymbol{\theta},i}(\boldsymbol{\epsilon})\big]} \,\middle|\, \boldsymbol{x},\boldsymbol{\theta}\right], \tag{24b}$$

where $\boldsymbol{s}$ and $\boldsymbol{s}^{\text{KSA}}$ equal each other up to leading orders in the width of the kernel. The derivation of (24) can be found in Appendix C.

## 3.3 Kernel Score Estimation using ML

We shall now explain how to use the kernel score approximation (24) to perform score estimation using machine learning. The method involves the following steps:

1. Choose a prior $\pi(\boldsymbol{\theta})$, kernel $K_{\boldsymbol{\theta}}(\boldsymbol{\epsilon})$, and difference function $\boldsymbol{\psi}_{\boldsymbol{x},\boldsymbol{\theta}}(\boldsymbol{\epsilon})$. For simplicity, the kernel $K_{\boldsymbol{\theta}}$ can be of the form

$$K_{\boldsymbol{\theta}}(\boldsymbol{\epsilon}) \equiv \left[\prod_{i=1}^d \frac{1}{\lambda_i(\boldsymbol{\theta})}\right] K(\boldsymbol{u}), \qquad \text{where } u_i = \frac{\epsilon_i}{\lambda_i(\boldsymbol{\theta})}. \tag{25}$$

Here $K$ is a $\boldsymbol{\theta}$-independent, standard-width, multi-dimensional kernel and $\lambda_i(\boldsymbol{\theta})$ are $\boldsymbol{\theta}$-dependent positive width parameters.

2. Sample datapoints $(\boldsymbol{x},\boldsymbol{\theta},\boldsymbol{\epsilon})$ from the distribution $\mathcal{P}$ in (23).

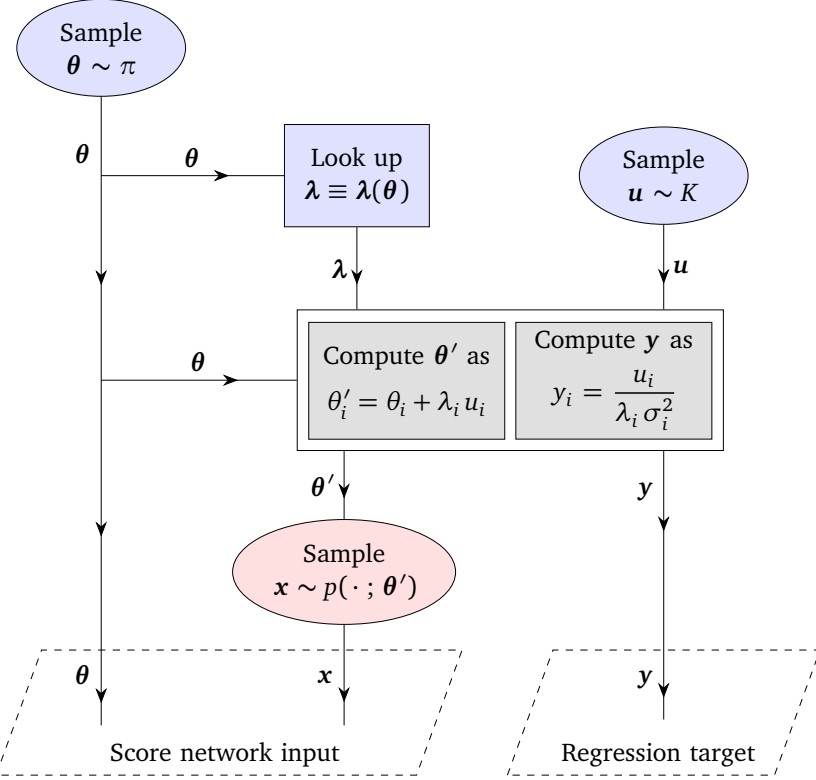

Figure 4: Flowchart depicting the generation of one datapoint $\big((x,\theta),y\big)$ for training the score network using KSA. Learning the score function can be treated as a regression problem, with $(x,\theta)$ as the network input and $y$ as the regression target. The $\psi_{x,\theta}^{(\mathrm{linear})}$ is used as the difference function, $\sigma_i^2$ is the variance of $u_i$ under the chosen standard-width kernel $K$.

3. Train a regression algorithm with $(x,\theta)$ serving as the input features and

$$\frac{\psi_{x,\theta,i}(\epsilon)}{\mathrm{E}_{\epsilon\sim K_\theta}\big[\epsilon_i\,\psi_{x,\theta,i}(\epsilon)\big]}$$

as the regression target. Note that with an appropriate choice for the kernel distribution $K_\theta(\epsilon)$ and the difference function $\psi_{x,\theta}(\epsilon)$, the denominator $\mathrm{E}_{\epsilon\sim K_\theta}\big[\epsilon_i\,\psi_{x,\theta,i}(\epsilon)\big]$ can be precomputed analytically.

4. Use the regressor trained in this manner as the estimated score function.

This procedure is illustrated in Figure 4 for the case where the difference function $\psi_{x,\theta}$ is linear as in (22) and the kernel $K_\theta(\epsilon)$ is of the form in (25).

   In Appendix D, we discuss the effect of some of the choices of KSE on the quality of the estimate $\hat{s}$. Based on certain criteria related to bias–variance trade-off, we recommend using i) the delta kernel in eq. (19) and ii) the linear difference function in (22) with KSE. The latter choice is already assumed in Figure 4, and the former can be incorporated by setting $K$ to

$$K^{\mathrm{delta}}(u) = \frac{1}{2^d}\prod_{i=1}^{d}\delta(|u_i|-1)\,. \tag{26}$$

Choosing $\lambda(\theta)$: In Appendix D, we also discuss the effect of the $d$-dimensional width parameter $\lambda(\theta)$. Larger widths increase the bias between the score $s$ and its approximation $s^{\mathrm{KSA}}$. On the other hand, larger widths make the regression problem of learning $s^{\mathrm{KSA}}$ easier by reducing the variance of the (components of the) regression target $y$ given a specific input $(x, \theta)$. The practitioner can choose the value of $\lambda(\theta)$ based on this bias–variance trade-off.

Note that if the standard-width kernel $K$ for sampling the $d$-dimensional $u$ in Figure 4 has a bounded support, say in $[-1, 1]^d$, then $y_i$ (and consequently $s_i^{\mathrm{KSA}}$) will be bounded by $\pm 1/(\lambda_i(\theta)\,\sigma_i^2)$. If the practitioner observes such a saturation in the estimated $\hat{s}_i$ values, it is a sign of high bias in $\hat{s}_i$. This can rectified by decreasing $\lambda_i$ for the relevant $\theta$ values.

## 3.4 Alternative Version of Kernel Score Approximation and Estimation

While (24) is the main result of this section, here we provide a modification that allows i) the width of the kernel to be dependent on $x$ (in addition to $\theta$), and ii) the attribute $x$ in the training dataset to be produced before the remaining attributes, namely $\theta$ and the regression target $y$.[4] Let the vector $\lambda(x, \theta)$ represent the positive width parameters of the kernel in each of the $d$ directions in the parameter space. Let $\odot$ and $\oslash$ represent the element-wise product and division operators between arrays, also known as Hadamard product and division operators:

$$(M \odot N)_i = M_i \cdot N_i, \qquad (M \oslash N)_i = M_i / N_i. \tag{27}$$

Now, we can write an $(x, \theta)$-dependent kernel $K_{x,\theta}$ in terms of $\lambda$ as

$$K_{x,\theta}(\epsilon) = \left[ \prod_{i=1}^{d} \frac{1}{\lambda_i(x, \theta)} \right] K\big(\epsilon \oslash \lambda(x, \theta)\big). \tag{28}$$

Let $\mathcal{Q}$ be the following unit-normalized joint-distribution of the triplet $(x, \theta, \epsilon)$:

$$\mathcal{Q}(x, \theta, \epsilon) \equiv \int d\theta'\, \pi(\theta') \cdot p(x\,;\theta') \cdot K_{x,\theta'}(\epsilon) \cdot \delta_{\mathrm{Dirac}}^{(d)}(\theta' - (\theta + \epsilon)) \tag{29a}$$

$$= \pi(\theta + \epsilon) \cdot K_{x,\theta+\epsilon}(\epsilon) \cdot p(x\,;\theta + \epsilon), \tag{29b}$$

where $\delta_{\mathrm{Dirac}}^{(d)}$ is the $d$-dimensional Dirac delta distribution. Introducing the weight

$$w(\theta, \epsilon) = \frac{\pi(\theta)}{\pi(\theta + \epsilon)}, \tag{30}$$

we can write the following alternative approximation $s^{\mathrm{KSA–alt}}$ for score function $s$:

$$\boxed{\begin{aligned} s_i(x\,&;\theta) \approx s_i^{\mathrm{KSA–alt}}(x\,;\theta) \\ &\equiv \frac{\mathrm{E}_{(x,\theta,\epsilon)\sim\mathcal{Q}}\left[ w(\theta, \epsilon)\, \dfrac{\lambda_i(x, \theta)}{\lambda_i(x, \theta + \epsilon)}\, \psi_{x,\theta,i}\big(\epsilon \odot \lambda(x, \theta) \oslash \lambda(x, \theta + \epsilon)\big) \,\Big|\, x, \theta \right]}{\mathrm{E}_{(x,\theta,\epsilon)\sim\mathcal{Q}}\left[ w(\theta, \epsilon) \,\big|\, x, \theta \right]\; \mathrm{E}_{\epsilon\sim K_{x,\theta}}\left[ \epsilon_i\, \psi_{x,\theta,i}(\epsilon) \right]}. \end{aligned}} \tag{31}$$

---

[4]Readers who are not interested in this case can safely skip Section 3.4.

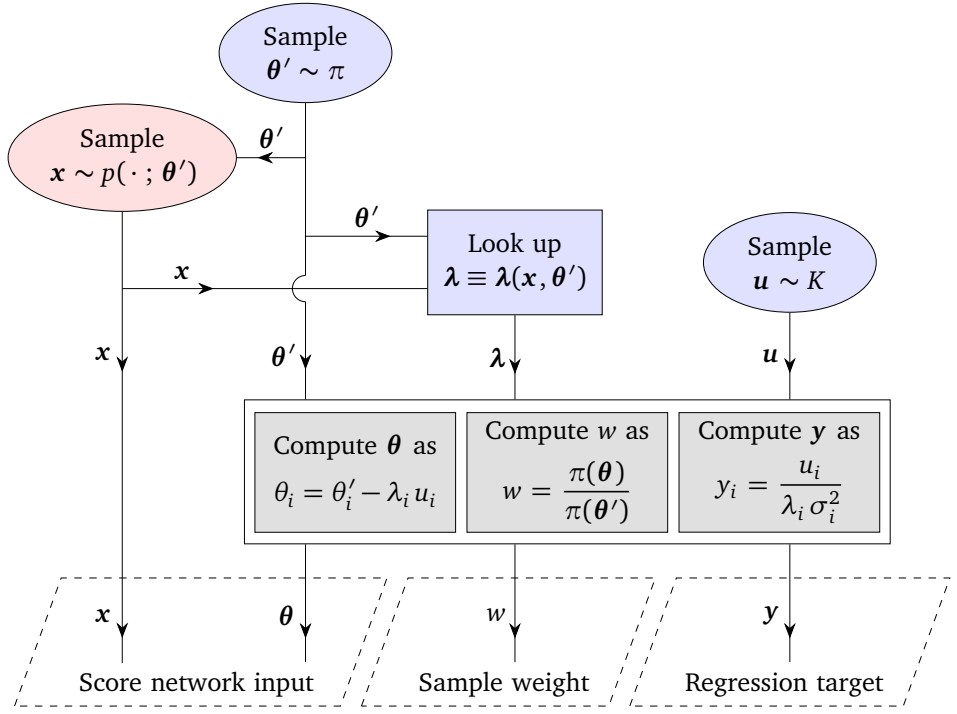

Figure 5: The same as Figure 4, but for training the the score network using the alternative version of KSA in (32). Learning the score function can be treated as a weighted regression problem, with $w$ being the sample weight.

For $\psi_{x,\theta,i}(\epsilon) = \psi_{x,\theta,i}^{(\text{linear})}(\epsilon) = \epsilon_i$, this becomes

$$s_i^{\text{KSA–alt}}(x\,;\,\theta) = \frac{\mathrm{E}_{(x,\theta,\epsilon)\sim\mathcal{Q}}\left[w(\theta,\epsilon)\,\dfrac{\lambda_i^2(x,\theta)}{\lambda_i^2(x,\theta+\epsilon)}\,\epsilon_i \,\bigg|\, x,\theta\right]}{\mathrm{E}_{(x,\theta,\epsilon)\sim\mathcal{Q}}\left[w(\theta,\epsilon)\,\big|\,x,\theta\right]}\;\frac{1}{\mathrm{E}_{\epsilon\sim K_{x,\theta}}\left[\epsilon_i^2\right]}. \tag{32}$$

One can use this approximation for the score function to perform the estimation of $s_i(x,\theta)$ as a weighted regression, with

$$\frac{\dfrac{\lambda_i^2(x,\theta)}{\lambda_i^2(x,\theta+\epsilon)}\,\epsilon_i}{\mathrm{E}_{\epsilon\sim K_{x,\theta}}\left[\epsilon_i^2\right]} \tag{33}$$

as the regression target, and $w(\theta,\epsilon) = \pi(\theta)/\pi(\theta+\epsilon)$ as the sample weight (the other basic steps are analogous to those described in Section 3.3). Figure 5 depicts this kernel score estimation procedure based on $s^{\text{KSA–alt}}$ in (32). The difference now is that a dataset of pairs $(\theta',x)$, which are computationally expensive to produce, can be prepared ahead of time, and reused multiple times in the subsequent steps of the pipeline in Figure 5. This allows for more efficient utilization of the produced events.

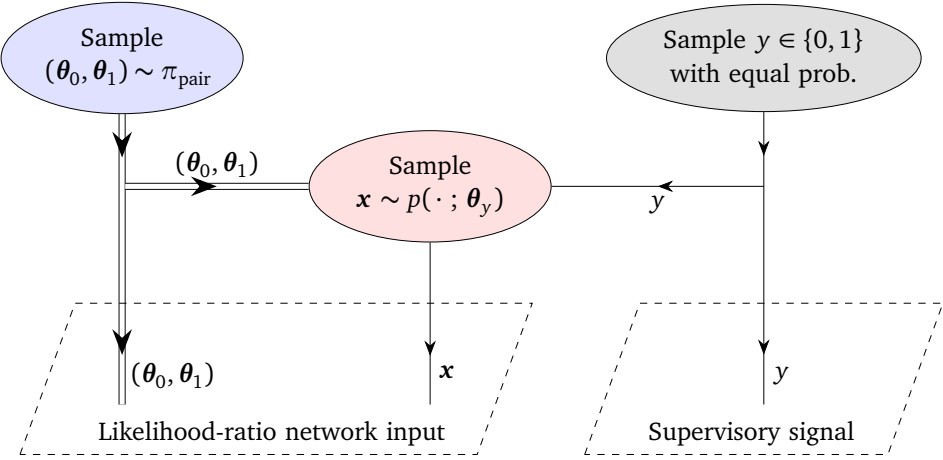

Figure 6: Flowchart depicting the generation of one datapoint $\big((\boldsymbol{x}, \boldsymbol{\theta}_0, \boldsymbol{\theta}_1), y\big)$ for training the likelihood ratio network. Learning the likelihood ratio function can be treated as a classification problem, with $(\boldsymbol{x}, \boldsymbol{\theta}_0, \boldsymbol{\theta}_1)$ as the network input and $y$ as the supervisory label.

## 4 Methodology: Kernel Likelihood Ratio Estimation

This section describes a technique (KLRE) for training a machine learning model to predict the doubly parameterized likelihood ratio $r(\boldsymbol{x}\,;\,\boldsymbol{\theta}_0, \boldsymbol{\theta}_1)$. It can be used with ISNs or any other NN architecture for modeling $\hat{r}$. This technique is comparatively easier to explain than the Kernel Score Estimation technique of Section 3. It is well known that, given two probability distributions (over the same data attributes), say $\mathcal{P}_0$ and $\mathcal{P}_1$, one can train a neural-network-based binary classifier to learn the likelihood ratio $\mathcal{P}_0/\mathcal{P}_1$ using labeled data from the two distributions, by minimizing certain loss functions known as *proper*[5] loss functions (e.g.: logistic loss, square loss, exponential loss, Savage loss, tangent loss). Here, we are interested in estimating the likelihood ratio $r(\boldsymbol{x}\,;\,\boldsymbol{\theta}_0, \boldsymbol{\theta}_1) = p(\boldsymbol{x}\,;\,\boldsymbol{\theta}_0)/p(\boldsymbol{x}\,;\,\boldsymbol{\theta}_1)$ for a range of values of $\boldsymbol{\theta}_0$ and $\boldsymbol{\theta}_1$, and not just between two specific distributions of $\boldsymbol{x}$. We can cast this task as a standard binary classification task by constructing joint-probability distributions $\mathcal{P}_0$ and $\mathcal{P}_1$ of the triplet $(\boldsymbol{x}, \boldsymbol{\theta}_0, \boldsymbol{\theta}_1)$ as follows:

$$\mathcal{P}_0(\boldsymbol{x}, \boldsymbol{\theta}_0, \boldsymbol{\theta}_1) \equiv \pi_{\mathrm{pair}}(\boldsymbol{\theta}_0, \boldsymbol{\theta}_1) \times p(\boldsymbol{x}\,;\,\boldsymbol{\theta}_0), \tag{34}$$
$$\mathcal{P}_1(\boldsymbol{x}, \boldsymbol{\theta}_0, \boldsymbol{\theta}_1) \equiv \pi_{\mathrm{pair}}(\boldsymbol{\theta}_0, \boldsymbol{\theta}_1) \times p(\boldsymbol{x}\,;\,\boldsymbol{\theta}_1).$$

Here $\pi_{\mathrm{pair}}$ is an *arbitrary* joint distribution of $(\boldsymbol{\theta}_0, \boldsymbol{\theta}_1)$. By this construction of $\mathcal{P}_y$, the doubly parameterized likelihood ratio function $r$ is simply $\mathcal{P}_0/\mathcal{P}_1$, since

$$\frac{\mathcal{P}_0(\boldsymbol{x}, \boldsymbol{\theta}_0, \boldsymbol{\theta}_1)}{\mathcal{P}_1(\boldsymbol{x}, \boldsymbol{\theta}_0, \boldsymbol{\theta}_1)} \equiv \frac{\cancel{\pi_{\mathrm{pair}}(\boldsymbol{\theta}_0, \boldsymbol{\theta}_1)} \times p(\boldsymbol{x}\,;\,\boldsymbol{\theta}_0)}{\cancel{\pi_{\mathrm{pair}}(\boldsymbol{\theta}_0, \boldsymbol{\theta}_1)} \times p(\boldsymbol{x}\,;\,\boldsymbol{\theta}_1)} \equiv r(\boldsymbol{x}\,;\,\boldsymbol{\theta}_0, \boldsymbol{\theta}_1). \tag{35}$$

To produce a datapoint $(\boldsymbol{x}, \boldsymbol{\theta}_0, \boldsymbol{\theta}_1)$ under $\mathcal{P}_y$, for $y = 0$ or 1, we sample $(\boldsymbol{\theta}_0, \boldsymbol{\theta}_1)$ as-per $\pi_{\mathrm{pair}}$ and sample an event $\boldsymbol{x}$ as-per $p(\boldsymbol{x}\,;\,\boldsymbol{\theta}_y)$. Figure 6 depicts this process for producing labeled datapoints from $\mathcal{P}_0$ and $\mathcal{P}_1$ (with equal probability). After producing training datasets this way, as mentioned

---

[5]These are loss functions for which (an estimate for) $\mathcal{P}_0/\mathcal{P}_1$ can be recovered from the trained neural network output.

previously, any *proper* binary classification loss function can be used to train the neural likelihood ratio function $\hat{r}$, using labeled data from $\mathcal{P}_0$ and $\mathcal{P}_1$. For completeness, here we provide some standard proper loss functions from the ML literature (for balanced classes, i.e., $y = 0$ or $1$ with equal probability), adapted for training the likelihood ratio estimate $\hat{r}$.

$$\mathcal{L}_{\text{logistic}}\left(\hat{r}, y\right) = -y \, \ln\left[\frac{1}{1+\hat{r}}\right] - (1-y) \ln\left[\frac{\hat{r}}{1+\hat{r}}\right], \tag{36a}$$

$$\mathcal{L}_{\text{square}}\left(\hat{r}, y\right) = \left[\frac{1}{1+\hat{r}} - y\right]^2, \tag{36b}$$

$$\mathcal{L}_{\text{exponential}}\left(\hat{r}, y\right) = y \, \sqrt{\hat{r}} + (1-y) \sqrt{\frac{1}{\hat{r}}}, \tag{36c}$$

$$\mathcal{L}_{\text{Savage}}\left(\hat{r}, y\right) = y \left(\frac{\hat{r}}{1+\hat{r}}\right)^2 + (1-y) \left(\frac{1}{1+\hat{r}}\right)^2. \tag{36d}$$

Special cases of this technique have appeared previously in the literature. In particular, the CARL technique for estimating $r$ uses independent and identically distributed $\boldsymbol{\theta}_0$ and $\boldsymbol{\theta}_1$:

$$\pi_{\text{pair}}^{\text{iid}}(\boldsymbol{\theta}_0, \boldsymbol{\theta}_1) \equiv \pi(\boldsymbol{\theta}_0)\,\pi(\boldsymbol{\theta}_1). \tag{37}$$

Likewise, the estimation of the singly-parameterized likelihood ratio $r_{\text{ref}}$ under the CARL and DCTR techniques can be interpreted as the special case where $\boldsymbol{\theta}_1$ is a constant, set to $\boldsymbol{\theta}_{\text{ref}}$:

$$\pi_{\text{pair}}^{\text{ref}}(\boldsymbol{\theta}_0, \boldsymbol{\theta}_1) \equiv \pi(\boldsymbol{\theta}_0)\,\delta_{\text{Dirac}}^{(d)}(\boldsymbol{\theta}_1 - \boldsymbol{\theta}_{\text{ref}}). \tag{38}$$

However, as discussed above, the likelihood ratio estimation technique is compatible with arbitrary distributions $\pi_{\text{pair}}$. In particular, we propose the use of symmetric correlated joint-distributions of the form

$$\pi_{\text{pair}}^{\text{kernel}}(\boldsymbol{\theta}_0, \boldsymbol{\theta}_1) = \frac{1}{2}\Big[\pi(\boldsymbol{\theta}_0)\,K_{\boldsymbol{\theta}_0}(\boldsymbol{\theta}_1 - \boldsymbol{\theta}_0) + \pi(\boldsymbol{\theta}_1)\,K_{\boldsymbol{\theta}_1}(\boldsymbol{\theta}_0 - \boldsymbol{\theta}_1)\Big], \tag{39}$$

where $K_{\boldsymbol{\theta}}$ is a $\boldsymbol{\theta}$ dependent kernel distribution localized around $\mathbf{0}$. We dub this the Kernel Likelihood Ratio Estimation (KLRE) technique. The performance of neural networks at different input values depends crucially on the distribution of the training data. Using the KLRE technique to train on correlated values of $\boldsymbol{\theta}_0$ and $\boldsymbol{\theta}_1$ can improve the performance of the neural network for neighboring parameter values.

Interestingly, **when using inferostatic networks** to model $\hat{r}$, due to the built-in symmetries of the network, if i) the loss function used in training is invariant under $y \mapsto (1-y); \hat{r} \mapsto 1/\hat{r}$, and ii) $\pi_{\text{pair}}$ is symmetric, then the training dataset need not be balanced, i.e., have the same proportions of $y = 0$ and $y = 1$, in order to use the loss functions in (36). More strongly, the proportions of training datapoints with $y = 0$ and $y = 1$ does not influence the training of the neural network.[6]

---

[6]Conversely, **when using ISNs**, if the training dataset is balanced, $\pi_{\text{pair}}$ will influence the training only through its symmetric part $[\pi_{\text{pair}}(\boldsymbol{\theta}_0, \boldsymbol{\theta}_1) + \pi_{\text{pair}}(\boldsymbol{\theta}_1, \boldsymbol{\theta}_0)]/2$.

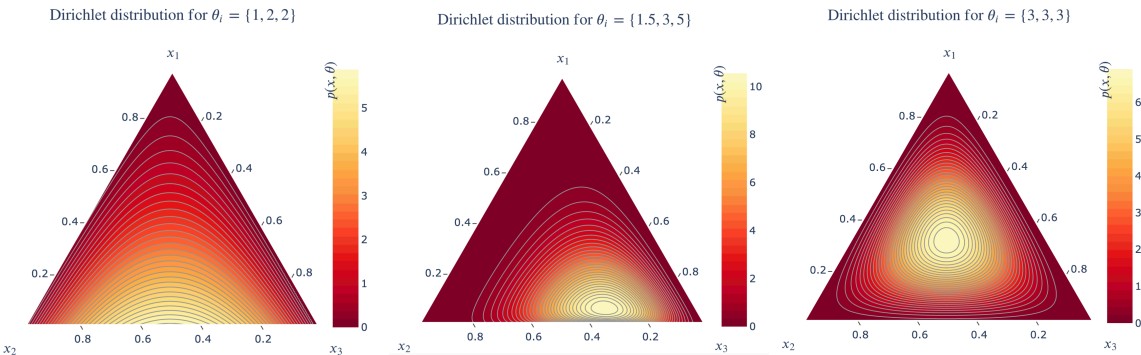

Figure 7: Illustration of the Dirichlet distribution (40) for $\boldsymbol{\theta} = (1,2,2)$ (left), $\boldsymbol{\theta} = (1.5,3,5)$ (middle) and $\boldsymbol{\theta} = (3,3,3)$ (right).

## 5 Experiments and Results

In this section, we demonstrate the various techniques introduced in this paper with an example. For our study, we use the 3-dimensional Dirichlet distribution given by

$$p(\boldsymbol{x} \,;\, \boldsymbol{\theta}) = \left[ \frac{\Gamma(\theta_1 + \theta_2 + \theta_3)}{\Gamma(\theta_1)\,\Gamma(\theta_2)\,\Gamma(\theta_3)} \right] \, x_1^{\theta_1 - 1}\, x_2^{\theta_2 - 1}\, x_3^{\theta_3 - 1}\,, \quad \text{where } \Gamma \text{ is the gamma function.} \tag{40}$$

This distribution i) has support over all 3-dimensional $\boldsymbol{x} \equiv (x_1, x_2, x_3)$ satisfying

$$x_1, x_2, x_3 \geq 0\,, \qquad x_1 + x_2 + x_3 = 1\,, \tag{41}$$

and ii) is parameterized by the 3-dimensional $\boldsymbol{\theta} \equiv (\theta_1, \theta_2, \theta_3)$ satisfying

$$\theta_1, \theta_2, \theta_3 > 0\,. \tag{42}$$

Figure 7 illustrates the 3-dimensional Dirichlet distribution for three different values of $\boldsymbol{\theta}$.

### 5.1 Tasks

In order to demonstrate our KSE and KLRE techniques, and compare it with the CARL technique, we will construct three tasks detailed below, one corresponding to each technique. For each task, we will train an ISN (which uses a backend $\hat{\varphi}$) and a "direct" NN (which directly models $\hat{s}$ and $\hat{r}$ as neural networks), using the corresponding training technique. In addition to evaluating the performance of the trained NNs in the tasks they were trained on, we will also evaluate them on tasks they were not trained on, but can nevertheless perform.

**Training details:** To accommodate for statistical variations in network performance, for each task we produce five training datasets of size 100,000. Five random initializations of an ISN and a direct NN are trained on these five training datasets. For each task, the details about the training loss function and the generation of the training data are provided below.

**Evaluation details:** For each task, we produce two evaluation datasets of size 100,000. All trained networks which can perform a given task will be evaluated on the same two datasets. The first testing dataset will be used to evaluate the average loss achieved by the networks. The second

testing dataset will be used to compare the neural network predictions against the corresponding true values (of the score or likelihood ratio) computed using our knowledge of the underlying distribution $p$. For each task, the details of the loss function and error metric used for these purposes are given below.

**Task 1: Kernel Score Estimation (KSE)**

Task 1 corresponds to the estimation of the score function using KSE. The training and evaluation datapoints $(x, \theta, y)$ are produced as-per the flowchart in Figure 4.

**Data generation details:** As already discussed in Section 3.3, in order to use our technique to estimate the score function, we make the following choices in Figure 4. For the "prior" distribution $\pi(\theta)$, we take an independent uniform prior for each $\theta_i$ between 0.5 and 5.

$$\pi(\theta) = \begin{cases} (5-0.5)^{-3}, & \text{if } 0.5 \leq \theta_1, \theta_2, \theta_3 < 5, \\ 0, & \text{otherwise}. \end{cases} \tag{43}$$

For the standard-width kernel $K$ used to generate $u$ in Figure 4, we use delta kernel in (26). For the width parameter $\lambda$ used to scale $u$, we use $\lambda_1(\theta) = \lambda_2(\theta) = \lambda_3(\theta) = 0.25$. Recall that for $\psi_{x,\theta}(\epsilon)$, the linear difference function from (22) is already chosen in Figure 4.

**Loss function:** In order to train and evaluate (using the first testing dataset) the NNs for this task, we use the mean-square-error as the per-datapoint loss function.

$$\mathcal{L}_{\text{mse}}(\hat{s}, y) = \frac{1}{3} \sum_{i=1}^{3} \left| \hat{s}_i(x \,;\, \theta) - y_i \right|^2. \tag{44}$$

**Error metric:** In order to evaluate (using the second testing dataset) the NNs for this task, we use the per-datapoint error metric given by

$$\mathcal{E}(\hat{s}, x, \theta) = \frac{1}{3} \sum_{i=1}^{3} \left| \hat{s}_i - s_i(x \,;\, \theta) \right|^2, \tag{45}$$

where $s(x \,;\, \theta)$ is the true score value for the given input $(x, \theta)$. For the 3-dimensional Dirichlet distribution in (40), it is given by

$$s_i(x \,;\, \theta) = \ln x_i + \text{digamma}(\theta_1 + \theta_2 + \theta_3) - \text{digamma}(\theta_i), \qquad \forall i \in \{1, 2, 3\}, \tag{46}$$

where digamma is the derivative of the natural logarithm of the gamma function.

**Task 2: Kernel Likelihood Ratio Estimation (KLRE)**

Task 2 corresponds to the estimation of the likelihood ratio function using KLRE. The training and evaluaton datapoints $(x, \theta_0, \theta_1, y)$ are produced as-per the flowchart in Figure 6.

**Data generation details:** We use the correlated joint-distribution $\pi_{\text{pair}}^{\text{kernel}}$ in (39) to produce $(\theta_0, \theta_1)$, with $\pi$ set to the prior in (43) and $K_\theta$ set to the rectangular kernel in (17) with constant width parameters $\lambda_1(\theta) = \lambda_2(\theta) = \lambda_3(\theta) = 0.4$.

**Loss function:** We train and evaluate (using the first testing dataset) the NNs for this task with the logistic loss for classification in (36a).

**Error metric:** We evaluate (using the second testing dataset) the NNs for this task using per-datapoint error metric given by

$$\mathcal{E}(\hat{r}, \boldsymbol{x}, \boldsymbol{\theta}_0, \boldsymbol{\theta}_1) = \left| \ln \hat{r} - \ln r(\boldsymbol{x} ; \boldsymbol{\theta}_0, \boldsymbol{\theta}_1) \right|^2, \tag{47}$$

where $r(\boldsymbol{x} ; \boldsymbol{\theta}_0, \boldsymbol{\theta}_1)$ is the true likelihood ratio value for the given input $\boldsymbol{x}, \boldsymbol{\theta}_0, \boldsymbol{\theta}_1$.

**Task 3: CARL**

Task 3 is identical to task 2, except that $\boldsymbol{\theta}_0$ and $\boldsymbol{\theta}_1$ are sampled independently (as-per $\pi_{\text{pair}}^{\text{iid}}$ in (37)), with $\pi$ set to the prior in (43).

## 5.2 NN Architecture and Training Details

There are a total of three different network architectures used in this study. The first is the ISN architecture, which models $\hat{\varphi}(\boldsymbol{x}, \boldsymbol{\theta})$, for the three ISN networks—one for each task. The second is a direct score network architecture, which models $\hat{s}(\boldsymbol{x} ; \boldsymbol{\theta}_0, \boldsymbol{\theta}_1)$ for the score estimation task (task 1). The third is a direct likelihood ratio network architecture, which models $\hat{r}(\boldsymbol{x} ; \boldsymbol{\theta}_0, \boldsymbol{\theta}_1)$ for tasks 2 and 3. For each of these cases, we use a simple feedforward neural network with three dense hidden layers with 8, 16, and 8 nodes, respectively. SELU [19] is used as the activation function in all the hidden layers. The networks only differ in their input and output specifications, which are provided in Table 2. Note that since shifting $\ln \hat{\varphi}$ by a constant does not affect $\hat{s}$ or $\hat{r}$ (and since the output layer of the ISN uses linear activation), we turn off the bias parameter in the output layer of the ISN network for $\ln \hat{\varphi}$. All the neural networks have a comparable number of trainable parameters (listed in the last column of Table 2).

As described earlier, for each task, five random initializations of an ISN network and a direct network are trained on five different training datasets, using the corresponding loss function. The study was performed using `TensorFlow-Keras` [20]. We trained the neural networks using the Adam optimizer[7] [21], for 20 epochs with a mini-batch size of 20 and using 10% of the training dataset as validation data. No noteworthy hyperparameter tuning was performed for training any of the networks. The wall time required for training was roughly the same for every combination of network architecture and training loss function.

## 5.3 Results

The results of this exercise are summarized in Table 3. For each task and each network that can perform that task, we show the median (over the five trained instances) of the average loss achieved by the network on the first testing dataset and the average error achieved by the network on the second testing dataset. The column headings indicate the technique used to train the network (KSE/KLRE/CARL) and the network architecture (ISN/"Dir" for direct). The row headings indicate the tasks the networks are being evaluated on. Note that networks trained using either KLRE or CARL can perform the tasks corresponding to both techniques. Furthermore, ISNs trained using any of the three techniques can perform all three tasks.

---

[7]With default settings in `Keras`, namely learning rate$= 0.001$, $\beta_1 = 0.9$, $\beta_2 = 0.999$, epsilon$= 10^{-7}$.

Table 2: Summary of the neural network architectures used in this paper.

| Network | Input | Output | Output layer | #params |
|---|---|---|---|---|
| ISN | $\boldsymbol{x}$ and $\boldsymbol{\theta}$ concatenated (6-dim) | $\ln \hat{\varphi}$ (1-dim) | Linear activation; No bias parameter | 344 |
| Direct score network | $\boldsymbol{x}$ and $\boldsymbol{\theta}$ concatenated (6-dim) | $\hat{\boldsymbol{s}}$ (3-dim) | Linear activation | 363 |
| Direct likelihood ratio network | $\boldsymbol{x}$, $\boldsymbol{\theta}_0$, and $\boldsymbol{\theta}_1$ concatenated (9-dim) | $(\zeta_0, \zeta_1)$ (2-dim) $\ln \hat{r} \equiv \zeta_0 - \zeta_1$ | Linear activation | 378 |

In order to visually compare the networks, for each task, in Figure 8 we show the average errors achieved (on the second testing dataset) by all five trained instances of all the networks that can perform the task. The networks are ordered along the $x$-axis in increasing order of the median (over the five network instances) average error—networks to the left are better. The error bars correspond to the uncertainty in the estimated average resulting from the finiteness of the testing dataset. From the results in Table 3 and Figure 8, we make the following observations:

1. **ISNs outperform their direct NN counterparts.** For each evaluation task, an ISN trained using a given training technique outperforms a direct network trained using the same technique (provided the direct network can perform the evaluation task). We attribute this to the **powerful and correct inductive bias**, incorporated into ISNs, regarding the nature of the function $\boldsymbol{s}$ (i.e., that it is the gradient of log-likelihood), and the function $r$ (i.e., that is is the ratio of likelihoods).

An important caveat here is that the performance of a network depends on a variety of factors including the size of training data, choice of training loss function (when several options are available), choice of evaluation metric, hyperparameter values, neural network architecture, and the likelihood function $p(\boldsymbol{x} \, ; \boldsymbol{\theta})$ under consideration.[8]

2. **Horses for courses.** Although an ISN trained using CARL or KLRE can be used to predict the score, we find that networks (ISN or direct) trained using the KSE technique show better performance as score predictors.

A direct network trained using KLRE performs poorly for the task corresponding to CARL. This is understandable, since there are no datapoints in KLRE training dataset with $\boldsymbol{\theta}_0 - \boldsymbol{\theta}_1$ outside a cube of length 0.8. A direct network lacks the structure needed to successfully extrapolate, from this limited training dataset, a good prediction of $r$ for independently sampled $\boldsymbol{\theta}_0$ and $\boldsymbol{\theta}_1$. Nevertheless, a direct network trained using KLRE outperforms the one trained using CARL for the KLRE task.

The ML training techniques introduced in this paper are not intended to be universally better than alternative techniques in the literature for all situations. Rather, as demonstrated here, different

---

[8]For a multivariate independent Gaussian distribution, with the mean-vector as the parameter $\boldsymbol{\theta}$, the score is a linear function of $\boldsymbol{x}$ and $\boldsymbol{\theta}$ whereas $\ln p$ is a quadratic function. This gives a direct score network an accidental advantage over an inferostatic score network, since dense feedforward architectures can model linear functions better than quadratic functions.

Table 3: The median evaluation-metric values obtained by first training and then independently evaluating on each of the three tasks described in the text (a total of 9 task-task pairings). In each case, we compare the result (test loss from (44) or (36a) and error from (45) or (47), respectively) from training an InferoStatic Network (ISN) or directly modeling (Dir.) the score as a fully connected NN as in [11]. The last column lists the result from using the true distribution (40).

| Eval. task | Performance metric | The task the machine was trained on | | | | | | True likelihood fn. |
|---|---|---|---|---|---|---|---|---|
| | | Task 1 (KSE) | | Task 2 (KLRE) | | Task 3 (CARL) | | |
| | | ISN | Dir. | ISN | Dir. | ISN | Dir. | |
| T–1 | Avg. loss | 15.515 | 15.540 | 15.695 | — | 15.594 | — | 15.515 |
| | Avg. error | 0.279 | 0.337 | 0.506 | — | 0.376 | — | 0 |
| T–2 | Avg. loss | 0.683 | — | 0.687 | 0.691 | 0.684 | 0.697 | 0.680 |
| | Avg. error | 0.049 | — | 0.082 | 0.120 | 0.063 | 0.177 | 0 |
| T–3 | Avg. loss | 0.431 | — | 0.472 | 0.690 | 0.424 | 0.442 | 0.415 |
| | Avg. error | 3.667 | — | 7.073 | 16.721 | 3.272 | 5.825 | 0 |

training techniques prioritize different aspects of the performance of the networks—the eventual use case of the trained neural network should guide the choice of training technique.

**3. Generalizability of ISNs.** We note that ISNs trained on one task generalize better for other tasks than the corresponding direct networks. A striking example of this is how, for the CARL-task, the ISNs trained using KLRE have comparable performance to the direct NNs trained using CARL. This is despite the fact that ISN-KLRE was trained only on datapoints with neighboring parameter values, and is possible because the ISN can efficiently extrapolate the value of $r$ for far-away $(\boldsymbol{\theta}_0, \boldsymbol{\theta}_1)$ values. Other demonstrations of the generalizability of ISNs include the relatively strong performance of KSE-ISN for tasks 2 and 3, and CARL-ISN for task 2 (KLRE). In summary, ISNs leverage built-in symmetries to generalize better to other tasks.

Finally, for completeness, we demonstrate the working of the two training techniques introduced in this paper, namely KSE and KLRE, by showing the heatmaps of the true vs predicted score functions (for the KSE-ISN with lowest average testing loss) in Figure 9, and the heatmap of the true vs predicted likelihood ratios (for the KLRE-ISN with lowest average testing loss) in Figure 10. Qualitatively, these figures show that the neural networks are indeed predicting score and likelihood functions, albeit imperfectly. The differences between the predicted and true values are due to limitations of the training process.

# 6 Conclusions and Outlook

In this work, we introduced InferoStatic Networks (ISN), a new architecture to model the score and likelihood ratio estimators in cases when the probability density can be sampled but not computed directly. The inferostatic potential is a scalar function that preserves many important features of likelihood ratios that are only approximately enforced in other approaches. ISNs can

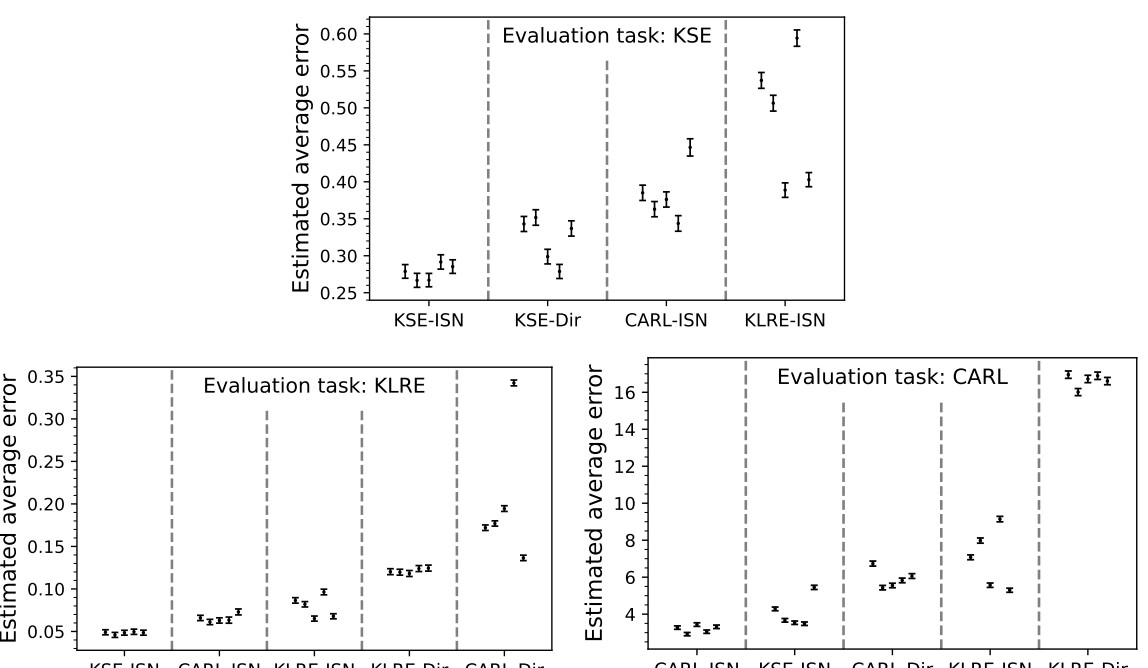

Figure 8: For each evaluation task, the estimated average errors for the five trained instances of the different NNs that can perform that task. The networks are ordered along the $x$-axis by median error. The error bars correspond to statistical uncertainty in the estimated average, from the finiteness of the testing dataset.

be used to learn both the score and the likelihood ratio simultaneously, while, in other approaches, these must be modelled by separate networks. The ISN also uses the available information more efficiently, which leads to faster, more accurate training. Since the fundamental properties (10) are automatically built-in, ISNs can better extrapolate into regions that are not well represented in the training set. We also automatically account for locality (neighboring points have similar functional values), which does not have to be learned from scratch by the network.

We also introduced the KSE technique, a new way to learn the score function, and the KLRE technique, which can learn the doubly parametrized likelihood ratio $r$. These techniques can be applied to the ISN or any other NN architecture, and do not require any latent information from a simulator. The KSE technique samples parameter values in the neighborhood of a probability distribution to estimate the derivative, and, hence, the score. The KLRE technique introduces correlations between parameters during training to learn composition properties of likelihood ratios. We also comment, but do not explore in detail, how latent information can be used more effectively (see Appendix A).

We have used a series of examples to demonstrate the advantages of the ISN technique and the KSE and KLRE methods. In all of these cases, the test loss is comparable to those of the standard methods, whereas the training error is reduced. We expect that these methods will be beneficial for multi-dimensional parameter fitting as encountered in event generator tuning.

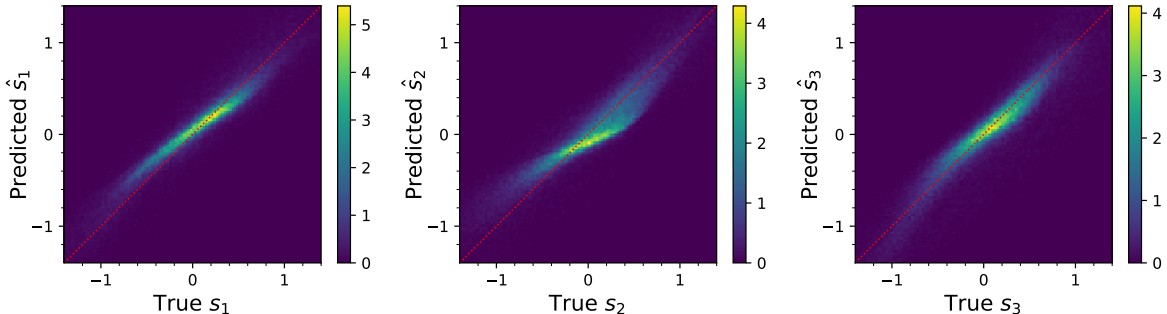

Figure 9: Density heatmaps showing the true score functions $s_i$ vs the predicted score functions $\hat{s}_i$, corresponding to the $i$-th component of the 3-dimensional parameter $\boldsymbol{\theta}$. The predictions are from the KSE-ISN with the lowest average testing loss, computed using testing dataset 1, for the KSE-task. The datapoints used in the heatmap are from testing dataset 2 of the KSE-task.

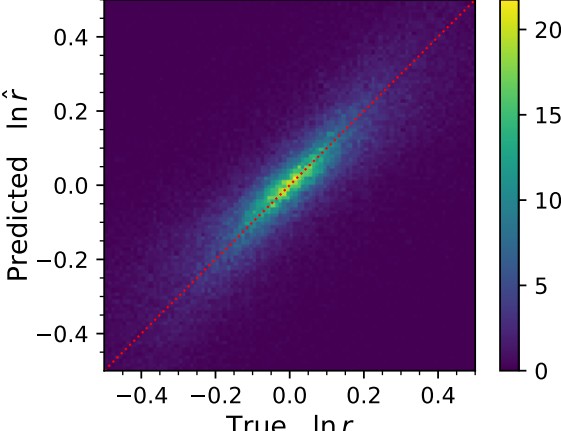

Figure 10: The same as Figure 9, but for KLRE-ISN network for task 2, i.e., the likelihood ratio estimation task corresponding to KLRE.

# Acknowledgements

The authors thank R. Houtz, A. Lee, and J. Thaler for useful discussions. Part of this work was performed at the Aspen Center for Physics, which is supported by National Science Foundation grant PHY-1607611. The authors would like to thank the Aspen Center for Physics for hospitality during the summer of 2022.

**Funding information**    This work is supported in parts by US DOE DE-SC0021447 and DOE DE-SC0022148. SM and PS are partially supported by the U.S. Department of Energy, Office of Science, Office of High Energy Physics QuantISED program under the grants "HEP Machine Learning and Optimization Go Quantum", Award Number 0000240323, and "DOE QuantiSED Consortium QCCFP-QMLQCF", Award Number DE-SC0019219. This manuscript has been authored by Fermi Research Alliance, LLC under Contract No. DEAC02-07CH11359 with the U.S. Department of

Energy, Office of Science, Office of High Energy Physics.

## Code and data availability

The code and data that support the findings of this study are openly available at the following URL: https://gitlab.com/prasanthcakewalk/code-and-data-availability/ under the directory named `arXiv_2210.01680`.

## A  New Loss Functions to Utilize Latent Information

### A.1  Background

The techniques discussed so far for estimating the score and likelihood ratio functions relied purely on being able to simulate the experimentally observable data-attributes under different theory models. However, a whole class of similar techniques exist that can use additional latent information from simulators when it is available [11]. In this section, we present new loss functions that can be used in those circumstances.

First, we review the use of latent information. Consider a latent attribute $z$ of the training data available from the simulator. Let $p_{\text{lat}}(x, z ; \theta)$ be the joint distribution of $(x, z)$ for a given $\theta$. The distribution of $p(x ; \theta)$ is given by

$$p(x ; \theta) \equiv \int dz\, p_{\text{lat}}(x, z ; \theta). \tag{48}$$

Assume that the quantity

$$r_{\text{lat}}(x, z ; \theta_0, \theta_1) \equiv \frac{p_{\text{lat}}(x, z ; \theta_0)}{p_{\text{lat}}(x, z ; \theta_1)} \tag{49}$$

can be calculated from the simulator.[9] In Ref. [11], it was shown how to use this latent information ("gold") extracted ("mined") from the simulator to improve the likelihood ratio estimation. The training proceeds exactly as the likelihood ratio estimation procedure outlined in Section 4, with two changes: i) an additional piece of information $r_{\text{lat}}$ is available as a supervisory signal, and ii) instead of standard classification losses, one of the following loss functions is used

$$\mathcal{L}_{\text{ROLR}}\left(\hat{r}, y, r_{\text{lat}}\right) = y\left(\hat{r} - r_{\text{lat}}\right)^2 + (1-y)\left(\frac{1}{\hat{r}} - \frac{1}{r_{\text{lat}}}\right)^2, \tag{50a}$$

$$\mathcal{L}_{\text{ALICE}}\left(\hat{r}, r_{\text{lat}}\right) = \frac{-1}{1 + r_{\text{lat}}} \ln\left[\frac{1}{1 + \hat{r}}\right] - \frac{r_{\text{lat}}}{1 + r_{\text{lat}}} \ln\left[\frac{\hat{r}}{1 + \hat{r}}\right]. \tag{50b}$$

The usage of $\mathcal{L}_{\text{ALICE}}$ is possible and beneficial because of the following relationships (proved in Appendix A.3):

$$\mathrm{E}_{\widetilde{\mathcal{P}}}\left[y \,\middle|\, x, \theta_0, \theta_1\right] = \mathrm{E}_{\widetilde{\mathcal{P}}}\left[\frac{1}{1 + r_{\text{lat}}} \,\middle|\, x, \theta_0, \theta_1\right], \tag{51}$$

---

[9]For example, if the simulation pipeline for producing $x$ given $\theta$ proceeds in two Markovian steps: produce a $z$ given $\theta$ and then produce an $x$ given $z$, i.e., $p(x ; \theta) = p_1(z ; \theta) p_2(x \,|\, z)$, then $r_{\text{lat}}(x, z ; \theta_0, \theta_1) = p_1(z ; \theta_0)/p_1(z ; \theta_1)$.

$$\text{Var}_{\widetilde{\mathcal{P}}}\left[y \;\middle|\; \boldsymbol{x}, \boldsymbol{\theta}_0, \boldsymbol{\theta}_1\right] \geq \text{Var}_{\widetilde{\mathcal{P}}}\left[\frac{1}{1 + r_{\text{lat}}} \;\middle|\; \boldsymbol{x}, \boldsymbol{\theta}_0, \boldsymbol{\theta}_1\right]. \tag{52}$$

This suggests that, when training a neural network with $(\boldsymbol{x}, \boldsymbol{\theta}_0, \boldsymbol{\theta}_1)$ as inputs using a loss function that is linear in $y$,[10] the training process can be improved by replacing $y$ with $1/(1 + r_{\text{lat}})$. This principle was used in Ref. [9,10] to derive $\mathcal{L}_{\text{ALICE}}$ from the logistic loss for classification. Extending the idea further, for any loss function $\mathcal{L}$, which uses $y$ and $r_{\text{lat}}$ as supervisory signals for training the likelihood ratio estimator $\hat{r}$, of the form

$$\mathcal{L}(\hat{r}, y, r_{\text{lat}}) = \mathcal{L}_1(\hat{r}, r_{\text{lat}}) + y \, \mathcal{L}_2(\hat{r}, r_{\text{lat}}) + \mathcal{L}_3(y, r_{\text{lat}}), \tag{53}$$

we can construct a corresponding loss function $\mathcal{L}_{\text{latent}}$ which uses only $r_{\text{lat}}$ as the supervisory signal as follows:

$$\mathcal{L}_{\text{latent}}(\hat{r}, r_{\text{lat}}) = \mathcal{L}_1(\hat{r}, r_{\text{lat}}) + \frac{1}{1 + r_{\text{lat}}} \mathcal{L}_2(\hat{r}, r_{\text{lat}}) + \mathcal{L}_3'(r_{\text{lat}}), \tag{54}$$

where $\mathcal{L}_1$ and $\mathcal{L}_2$ are the same functions used in (53) and $\mathcal{L}_3'$ is an arbitrary real-valued function of $r_{\text{lat}}$. Let $w$ be any one of the trainable parameters of the neural network function $\hat{r}$. We show in Appendix A.3 that[11]

$$\text{E}_{\widetilde{\mathcal{P}}}\left[\frac{\partial \, \mathcal{L}_{\text{latent}}(\hat{r}, r_{\text{lat}})}{\partial \, w}\bigg|_{\hat{r}=\hat{r}(\boldsymbol{x}, \boldsymbol{\theta}_0, \boldsymbol{\theta}_1)}\right] = \text{E}_{\widetilde{\mathcal{P}}}\left[\frac{\partial \, \mathcal{L}(\hat{r}, y, r_{\text{lat}})}{\partial \, w}\bigg|_{\hat{r}=\hat{r}(\boldsymbol{x}, \boldsymbol{\theta}_0, \boldsymbol{\theta}_1)}\right], \tag{55}$$

$$\text{Var}_{\widetilde{\mathcal{P}}}\left[\frac{\partial \, \mathcal{L}_{\text{latent}}(\hat{r}, r_{\text{lat}})}{\partial \, w}\bigg|_{\hat{r}=\hat{r}(\boldsymbol{x}, \boldsymbol{\theta}_0, \boldsymbol{\theta}_1)}\right] \leq \text{Var}_{\widetilde{\mathcal{P}}}\left[\frac{\partial \, \mathcal{L}(\hat{r}, y, r_{\text{lat}})}{\partial \, w}\bigg|_{\hat{r}=\hat{r}(\boldsymbol{x}, \boldsymbol{\theta}_0, \boldsymbol{\theta}_1)}\right]. \tag{56}$$

This means that $\mathcal{L}_{\text{latent}}$ has the same average derivative with respect to $w$ as $\mathcal{L}$, but has a lower variance in the derivative. This can lead to a more data-efficient a) estimation of the average gradient, and consequently b) training of the neural network parameters.

## A.2   New Loss Functions

Using the construction in (54), we provide the following loss functions for training the function $\hat{r}$ using $r_{\text{lat}}$ as the supervisory signal:

$$\mathcal{L}_{\text{latent\_ROLR}}\left(\hat{r}, r_{\text{lat}}\right) = \frac{1}{1 + r_{\text{lat}}}\left(\hat{r} - r_{\text{lat}}\right)^2 + \frac{r_{\text{lat}}}{1 + r_{\text{lat}}}\left(\frac{1}{\hat{r}} - \frac{1}{r_{\text{lat}}}\right)^2, \tag{57a}$$

$$\mathcal{L}_{\text{latent\_square}}\left(\hat{r}, r_{\text{lat}}\right) = \left[\frac{1}{1 + \hat{r}} - \frac{1}{1 + r_{\text{lat}}}\right]^2 = \frac{\left(\hat{r} - r_{\text{lat}}\right)^2}{\left(1 + r_{\text{lat}}\right)^2\left(1 + \hat{r}\right)^2}, \tag{57b}$$

$$\mathcal{L}_{\text{latent\_exponential}}\left(\hat{r}, r_{\text{lat}}\right) = \frac{1}{1 + r_{\text{lat}}}\sqrt{\hat{r}} + \frac{r_{\text{lat}}}{1 + r_{\text{lat}}}\sqrt{\frac{1}{\hat{r}}}, \tag{57c}$$

$$\mathcal{L}_{\text{latent\_Savage}}\left(\hat{r}, r_{\text{lat}}\right) = \frac{1}{1 + r_{\text{lat}}}\left(\frac{\hat{r}}{1 + \hat{r}}\right)^2 + \frac{r_{\text{lat}}}{1 + r_{\text{lat}}}\left(\frac{1}{1 + \hat{r}}\right)^2. \tag{57d}$$

---

[10]Note that since $y$ is a binary variable, any real-valued function of $y$ can be equivalently written as a linear function of $y$.

[11]The dependence of $\hat{r}$ on the NN weights is left implicit.

They are, in order, the low-variance versions of $\mathcal{L}_{\text{ROLR}}$ in (50a), and square, exponential, and Savage losses from (36). Using the construction in (54) on the logistic loss leads to $\mathcal{L}_{\text{ALICE}}$ in (50b) from Ref. [10].

### A.3 Proofs

**Proof of** (51)

We will prove a stronger version of (51) here. Let $f$ and $g$ be arbitrary real-valued functions of $(r_{\text{lat}}, \boldsymbol{x}, \boldsymbol{\theta}_0, \boldsymbol{\theta}_1)$. Using the law of total expectation, we can write

$$\mathrm{E}_{\widetilde{\mathcal{P}}}\left[ y f + g \;\middle|\; \boldsymbol{x}, \boldsymbol{\theta}_0, \boldsymbol{\theta}_1 \right] = \mathrm{E}_{\widetilde{\mathcal{P}}}\left[ \mathrm{E}_{\widetilde{\mathcal{P}}}\left[ y f + g \;\middle|\; \boldsymbol{x}, \boldsymbol{\theta}_0, \boldsymbol{\theta}_1, \boldsymbol{z} \right] \middle|\; \boldsymbol{x}, \boldsymbol{\theta}_0, \boldsymbol{\theta}_1 \right]. \tag{58}$$

Since, $r_{\text{lat}}$ is completely determined by $(\boldsymbol{x}, \boldsymbol{z}, \boldsymbol{\theta}_0, \boldsymbol{\theta}_1)$, it follows that $f$ and $g$ are also completely determined by $(\boldsymbol{x}, \boldsymbol{z}, \boldsymbol{\theta}_0, \boldsymbol{\theta}_1)$. This leads to

$$\mathrm{E}_{\widetilde{\mathcal{P}}}\left[ y f + g \;\middle|\; \boldsymbol{x}, \boldsymbol{\theta}_0, \boldsymbol{\theta}_1 \right] = \mathrm{E}_{\widetilde{\mathcal{P}}}\left[ \mathrm{E}_{\widetilde{\mathcal{P}}}\left[ y \;\middle|\; \boldsymbol{x}, \boldsymbol{\theta}_0, \boldsymbol{\theta}_1, \boldsymbol{z} \right] f + g \;\middle|\; \boldsymbol{x}, \boldsymbol{\theta}_0, \boldsymbol{\theta}_1 \right]. \tag{59}$$

From the definition of $r_{\text{lat}}$ in (49), this simplifies to

$$\mathrm{E}_{\widetilde{\mathcal{P}}}\left[ y f + g \;\middle|\; \boldsymbol{x}, \boldsymbol{\theta}_0, \boldsymbol{\theta}_1 \right] = \mathrm{E}_{\widetilde{\mathcal{P}}}\left[ \left( \frac{1}{1 + r_{\text{lat}}} \right) f + g \;\middle|\; \boldsymbol{x}, \boldsymbol{\theta}_0, \boldsymbol{\theta}_1 \right]. \tag{60}$$

Equation (51) is a special case of this result for $f(r_{\text{lat}}, \boldsymbol{x}, \boldsymbol{\theta}_0, \boldsymbol{\theta}_1) \equiv 1$ and $g(r_{\text{lat}}, \boldsymbol{x}, \boldsymbol{\theta}_0, \boldsymbol{\theta}_1) \equiv 0$.

**Proof of** (52)

As before, we will prove a stronger version of (52) here, for arbitrary functions $f$ and $g$ described above. Using the law of total variance, we can write

$$\mathrm{Var}_{\widetilde{\mathcal{P}}}\left[ y f + g \;\middle|\; \boldsymbol{x}, \boldsymbol{\theta}_0, \boldsymbol{\theta}_1 \right] = \mathrm{Var}_{\widetilde{\mathcal{P}}}\left[ \mathrm{E}_{\widetilde{\mathcal{P}}}\left[ y f + g \;\middle|\; \boldsymbol{x}, \boldsymbol{\theta}_0, \boldsymbol{\theta}_1, \boldsymbol{z} \right] \middle|\; \boldsymbol{x}, \boldsymbol{\theta}_0, \boldsymbol{\theta}_1 \right]$$
$$+ \mathrm{E}_{\widetilde{\mathcal{P}}}\left[ \mathrm{Var}_{\widetilde{\mathcal{P}}}\left[ y f + g \;\middle|\; \boldsymbol{x}, \boldsymbol{\theta}_0, \boldsymbol{\theta}_1, \boldsymbol{z} \right] \middle|\; \boldsymbol{x}, \boldsymbol{\theta}_0, \boldsymbol{\theta}_1 \right]. \tag{61}$$

Using the facts that $f$ and $g$ are completely determined by $(\boldsymbol{x}, \boldsymbol{z}, \boldsymbol{\theta}_0, \boldsymbol{\theta}_1)$, and the definition of $r_{\text{lat}}$, this can be written as

$$\mathrm{Var}_{\widetilde{\mathcal{P}}}\left[ y f + g \;\middle|\; \boldsymbol{x}, \boldsymbol{\theta}_0, \boldsymbol{\theta}_1 \right] = \mathrm{Var}_{\widetilde{\mathcal{P}}}\left[ \left( \frac{1}{1 + r_{\text{lat}}} \right) f + g \;\middle|\; \boldsymbol{x}, \boldsymbol{\theta}_0, \boldsymbol{\theta}_1 \right]$$
$$+ \mathrm{E}_{\widetilde{\mathcal{P}}}\left[ \mathrm{Var}_{\widetilde{\mathcal{P}}}\left[ y f + g \;\middle|\; \boldsymbol{x}, \boldsymbol{\theta}_0, \boldsymbol{\theta}_1, \boldsymbol{z} \right] \middle|\; \boldsymbol{x}, \boldsymbol{\theta}_0, \boldsymbol{\theta}_1 \right]. \tag{62}$$

From the non-negativity of variances, it follows that

$$\mathrm{Var}_{\widetilde{\mathcal{P}}}\left[ y f + g \;\middle|\; \boldsymbol{x}, \boldsymbol{\theta}_0, \boldsymbol{\theta}_1 \right] \geq \mathrm{Var}_{\widetilde{\mathcal{P}}}\left[ \left( \frac{1}{1 + r_{\text{lat}}} \right) f + g \;\middle|\; \boldsymbol{x}, \boldsymbol{\theta}_0, \boldsymbol{\theta}_1 \right]. \tag{63}$$

Equation (52) is a special case of this result for $f(r_{\text{lat}}, \boldsymbol{x}, \boldsymbol{\theta}_0, \boldsymbol{\theta}_1) \equiv 1$ and $g(r_{\text{lat}}, \boldsymbol{x}, \boldsymbol{\theta}_0, \boldsymbol{\theta}_1) \equiv 0$.

**Proof of** (55)

Using the law of total expectation, and leaving implicit the dependence of $\hat{r}$ on $\boldsymbol{x}$, $\boldsymbol{\theta}_0$, $\boldsymbol{\theta}_1$, and $w$, the right hand side of (55) can be written as

$$
\mathrm{E}_{\widetilde{\mathcal{P}}}\left[\frac{\partial\,\mathcal{L}(\hat{r},y,r_{\mathrm{lat}})}{\partial\,w}\right]=\mathrm{E}_{\widetilde{\mathcal{P}}}\left[\mathrm{E}_{\widetilde{\mathcal{P}}}\left[\frac{\partial\,\mathcal{L}(\hat{r},y,r_{\mathrm{lat}})}{\partial\,\hat{r}}\,\frac{\partial\,\hat{r}}{\partial\,w}\,\bigg|\,\boldsymbol{x},\boldsymbol{\theta}_0,\boldsymbol{\theta}_1\right]\right]. \tag{64}
$$

From the form of $\mathcal{L}$ in (53), this can be written as

$$
\mathrm{E}_{\widetilde{\mathcal{P}}}\left[\frac{\partial\,\mathcal{L}(\hat{r},y,r_{\mathrm{lat}})}{\partial\,w}\right]=\mathrm{E}_{\widetilde{\mathcal{P}}}\left[\mathrm{E}_{\widetilde{\mathcal{P}}}\left[\left(\frac{\partial\,\mathcal{L}_1(\hat{r},r_{\mathrm{lat}})}{\partial\,\hat{r}}+y\,\frac{\partial\,\mathcal{L}_2(\hat{r},r_{\mathrm{lat}})}{\partial\,\hat{r}}\right)\frac{\partial\,\hat{r}}{\partial\,w}\,\bigg|\,\boldsymbol{x},\boldsymbol{\theta}_0,\boldsymbol{\theta}_1\right]\right]. \tag{65}
$$

Likewise, the left hand side of (55) can be written as

$$
\mathrm{E}_{\widetilde{\mathcal{P}}}\left[\frac{\partial\,\mathcal{L}_{\mathrm{latent}}(\hat{r},r_{\mathrm{lat}})}{\partial\,w}\right]=\mathrm{E}_{\widetilde{\mathcal{P}}}\left[\mathrm{E}_{\widetilde{\mathcal{P}}}\left[\left(\frac{\partial\,\mathcal{L}_1(\hat{r},r_{\mathrm{lat}})}{\partial\,\hat{r}}+\frac{1}{1+r_{\mathrm{lat}}}\,\frac{\partial\,\mathcal{L}_2(\hat{r},r_{\mathrm{lat}})}{\partial\,\hat{r}}\right)\frac{\partial\,\hat{r}}{\partial\,w}\,\bigg|\,\boldsymbol{x},\boldsymbol{\theta}_0,\boldsymbol{\theta}_1\right]\right]. \tag{66}
$$

The equality of the two sides of (55) now follows from the result in (60), with the following choice for the functions $f$ and $g$:

$$
f(r_{\mathrm{lat}},\boldsymbol{x},\boldsymbol{\theta}_0,\boldsymbol{\theta}_1)\equiv\frac{\partial\,\mathcal{L}_2(\hat{r},r_{\mathrm{lat}})}{\partial\,\hat{r}}\,\frac{\partial\,\hat{r}}{\partial\,w}\,, \tag{67a}
$$

$$
g(r_{\mathrm{lat}},\boldsymbol{x},\boldsymbol{\theta}_0,\boldsymbol{\theta}_1)\equiv\frac{\partial\,\mathcal{L}_1(\hat{r},r_{\mathrm{lat}})}{\partial\,\hat{r}}\,\frac{\partial\,\hat{r}}{\partial\,w}\,. \tag{67b}
$$

**Proof of** (56)

Using the law of total variance, we can write the left hand side of (56) as

$$
\mathrm{Var}_{\widetilde{\mathcal{P}}}\left[\frac{\partial\,\mathcal{L}_{\mathrm{latent}}(\hat{r},r_{\mathrm{lat}})}{\partial\,w}\right]=\mathrm{Var}_{\widetilde{\mathcal{P}}}\left[\mathrm{E}_{\widetilde{\mathcal{P}}}\left[\left(\frac{\partial\,\mathcal{L}_1(\hat{r},r_{\mathrm{lat}})}{\partial\,\hat{r}}+\frac{1}{1+r_{\mathrm{lat}}}\,\frac{\partial\,\mathcal{L}_2(\hat{r},r_{\mathrm{lat}})}{\partial\,\hat{r}}\right)\frac{\partial\,\hat{r}}{\partial\,w}\,\bigg|\,\boldsymbol{x},\boldsymbol{\theta}_0,\boldsymbol{\theta}_1\right]\right]
$$
$$
+\,\mathrm{E}_{\widetilde{\mathcal{P}}}\left[\mathrm{Var}_{\widetilde{\mathcal{P}}}\left[\left(\frac{\partial\,\mathcal{L}_1(\hat{r},r_{\mathrm{lat}})}{\partial\,\hat{r}}+\frac{1}{1+r_{\mathrm{lat}}}\,\frac{\partial\,\mathcal{L}_2(\hat{r},r_{\mathrm{lat}})}{\partial\,\hat{r}}\right)\frac{\partial\,\hat{r}}{\partial\,w}\,\bigg|\,\boldsymbol{x},\boldsymbol{\theta}_0,\boldsymbol{\theta}_1\right]\right], \tag{68}
$$

and the right hand side of (56) as

$$
\mathrm{Var}_{\widetilde{\mathcal{P}}}\left[\frac{\partial\,\mathcal{L}(\hat{r},y,r_{\mathrm{lat}})}{\partial\,w}\right]=\mathrm{Var}_{\widetilde{\mathcal{P}}}\left[\mathrm{E}_{\widetilde{\mathcal{P}}}\left[\left(\frac{\partial\,\mathcal{L}_1(\hat{r},r_{\mathrm{lat}})}{\partial\,\hat{r}}+y\,\frac{\partial\,\mathcal{L}_2(\hat{r},r_{\mathrm{lat}})}{\partial\,\hat{r}}\right)\frac{\partial\,\hat{r}}{\partial\,w}\,\bigg|\,\boldsymbol{x},\boldsymbol{\theta}_0,\boldsymbol{\theta}_1\right]\right]
$$
$$
+\,\mathrm{E}_{\widetilde{\mathcal{P}}}\left[\mathrm{Var}_{\widetilde{\mathcal{P}}}\left[\left(\frac{\partial\,\mathcal{L}_1(\hat{r},r_{\mathrm{lat}})}{\partial\,\hat{r}}+y\,\frac{\partial\,\mathcal{L}_2(\hat{r},r_{\mathrm{lat}})}{\partial\,\hat{r}}\right)\frac{\partial\,\hat{r}}{\partial\,w}\,\bigg|\,\boldsymbol{x},\boldsymbol{\theta}_0,\boldsymbol{\theta}_1\right]\right]. \tag{69}
$$

Using the results in (60) and (63) with the choice for $f$ and $g$ given in (67), it follows that

$$
\mathrm{Var}_{\widetilde{\mathcal{P}}}\left[\frac{\partial\,\mathcal{L}_{\mathrm{latent}}(\hat{r},r_{\mathrm{lat}})}{\partial\,w}\right]\leq\mathrm{Var}_{\widetilde{\mathcal{P}}}\left[\frac{\partial\,\mathcal{L}(\hat{r},y,r_{\mathrm{lat}})}{\partial\,w}\right], \tag{70}
$$

which completes the proof of (56).

# B Feed-Forward Nature of the Gradient Network

In this section, we show that if the scalar function $\hat{\varphi}(x, \theta)$ is modeled as a feed-forward neural network, then its gradient $\hat{s}(x, \theta) \equiv \nabla_\theta \hat{\varphi}(x, \theta)$ can also be expressed as a feed-forward network, with the exact same trainable neural network weights or parameters. Let us begin with the description of a generic feed-forward network modeling $\hat{\varphi}$. For simplicity, we will consider each layer of the network in its flattened form. In other words, the input layer, output layer, and the intermediate layers of the network are all vectors (indexed from 1). Let the input layer be the vector $a^{(0)}$ given by

$$a^{(0)} \equiv (\theta, x), \tag{71}$$

$$a_i^{(0)} \equiv \begin{cases} \theta_i, & \text{if } 1 \leq i \leq d, \\ x_{i-d}, & \text{if } i > d, \end{cases} \tag{72}$$

where $d = \dim(\theta)$ is the dimensionality of $\theta$. The subsequent layers of the neural network are of the form

$$a^{(n)} \equiv f^{(n)}\left(a^{(0)}, \ldots, a^{(n-1)}\right), \qquad \forall n = 1, \ldots, N, \tag{73}$$

where $f^{(n)}$ is a trainable, $d_n$-dimensional function which depends only on $a^{(0)}, \ldots, a^{(n-1)}$. The fact that each layer only depends on the layers that precede it makes this a feed-forward network. The value of the function $\hat{\varphi}$ is identified with the final ($N$-th) layer of the network, which is chosen to be one dimensional, i.e., $d_N = 1$:

$$\hat{\varphi} \equiv a_1^{(N)}. \tag{74}$$

Now let us move onto the description of the network for the gradient function $\hat{s}$. Let $B^{(n)}$ be the gradient of the vector layer $a^{(n)}$ with respect to $\theta$. $B^{(n)}$ is a $d_n \times d$ dimensional matrix given by

$$B^{(n)} \equiv \nabla_\theta a^{(n)}, \qquad \forall n = 0, \ldots, N, \tag{75}$$

$$B_{ij}^{(n)} \equiv \frac{\partial a_i^{(n)}}{\partial \theta_j}, \qquad \forall n = 0, \ldots, N. \tag{76}$$

As we will see, $\left\{a^{(n)}\right\}_{n=0}^N$ and $\left\{B^{(n)}\right\}_{n=0}^N$ together form the layers of the gradient network for computing $\hat{s}$. From (72), we can see that $B^{(0)}$ is simply a constant matrix given by

$$B_{ij}^{(0)} = \begin{cases} 1, & \text{if } 1 \leq i = j \leq d, \\ 0, & \text{otherwise}. \end{cases} \tag{77}$$

Subsequent $B^{(n)}$-s for $n = 1, \ldots, N$ are given by

$$B_{ij}^{(n)} = \frac{\partial a_i^{(n)}}{\partial \theta_j} = \sum_{m=0}^{n-1} \sum_{k=1}^{d_m} \frac{\partial f_i^{(n)}}{\partial a_k^{(m)}} \cdot \frac{\partial a_k^{(m)}}{\partial \theta_j} = \sum_{m=0}^{n-1} \sum_{k=1}^{d_m} \frac{\partial f_i^{(n)}}{\partial a_k^{(m)}} \cdot B_{kj}^{(m)}. \tag{78}$$

Note that each $\partial f_i^{(n)} / \partial a_k^{(m)}$ in this expression is a) simply a function of $a^{(0)}, \ldots, a^{(n-1)}$, and b) has the same trainable parameters as the function $f^{(n)}$. This means that $B^{(n)}$ can be written as

$$B^{(n)} \equiv G^{(n)}\left(a^{(0)}, \ldots, a^{(n-1)}, B^{(0)}, \ldots, B^{(n-1)}\right), \tag{79}$$

where $\boldsymbol{G}^{(n)}$ is a $d_n \times d$ dimensional function with the exact same trainable parameters as $\boldsymbol{f}^{(n)}$. This means that for $n \in \{1, \ldots, N\}$, $\boldsymbol{a}^{(n)}$ and $\boldsymbol{B}^{(n)}$ can computed in a feed-forward manner using $\boldsymbol{f}^{(n)}$ and $\boldsymbol{G}^{(n)}$. The score function to be computed by the gradient network can be identified with the final layer $\boldsymbol{B}^{(N)}$, since

$$\hat{s}_i \equiv \frac{\partial \hat{\varphi}}{\partial \theta_i} \equiv \frac{\partial a_1^{(N)}}{\partial \theta_i} \equiv B_{1i}^{(N)} . \tag{80}$$

This shows that the gradient network for $\hat{\boldsymbol{s}}$ is also feed-forward in nature. Furthermore, the gradient network will have the same dependency graph between layers as the original network for $\hat{\varphi}$. This can be inferred from (78)—if $\boldsymbol{f}^{(n)}$ does not directly depend on $\boldsymbol{a}^{(m)}$ for some $m < n$, then $\partial f_i^{(n)} / \partial a_k^{(m)} \equiv 0$ for all $(i, k)$, which in turn implies that $\boldsymbol{G}^{(n)}$ does not directly depend on $\boldsymbol{a}^{(m)}$ or $\boldsymbol{B}^{(m)}$.

An important consequence of this proof is that if the ISN for $\hat{\varphi}(\boldsymbol{x}, \boldsymbol{\theta})$ is modeled as a feed-forward neural network, then its gradient (i.e., the score network) can be trained using backpropagation.

## C Derivation of the Kernel Score Estimation Technique

The kernel distribution $K_{\boldsymbol{\theta}}$ which satisfies (16) and the difference function $\boldsymbol{\psi}_{\boldsymbol{x},\boldsymbol{\theta}}$ which satisfies (20), together obey the following properties:

$$\int d\boldsymbol{\epsilon} \, K_{\boldsymbol{\theta}}(\boldsymbol{\epsilon}) = 1 , \tag{81a}$$

$$\int d\boldsymbol{\epsilon} \, K_{\boldsymbol{\theta}}(\boldsymbol{\epsilon}) \, \epsilon_i = \int d\boldsymbol{\epsilon} \, K_{\boldsymbol{\theta}}(\boldsymbol{\epsilon}) \, \psi_{\boldsymbol{x},\boldsymbol{\theta},i}(\boldsymbol{\epsilon}) = 0 , \qquad \forall i , \tag{81b}$$

$$\int d\boldsymbol{\epsilon} \, K_{\boldsymbol{\theta}}(\boldsymbol{\epsilon}) \, \epsilon_i \, \epsilon_j = \int d\boldsymbol{\epsilon} \, K_{\boldsymbol{\theta}}(\boldsymbol{\epsilon}) \, \psi_{\boldsymbol{x},\boldsymbol{\theta},i}(\boldsymbol{\epsilon}) \, \epsilon_j = 0 , \qquad \forall i \neq j , \tag{81c}$$

$$\int d\boldsymbol{\epsilon} \, K_{\boldsymbol{\theta}}(\boldsymbol{\epsilon}) \, \psi_{\boldsymbol{x},\boldsymbol{\theta},i}(\boldsymbol{\epsilon}) \, \epsilon_j \, \epsilon_k = 0 , \qquad \forall i, j, k . \tag{81d}$$

For small $\boldsymbol{\epsilon}$, $p(\boldsymbol{x} ; \boldsymbol{\theta} + \boldsymbol{\epsilon})$ can be approximated as

$$\begin{aligned}
p(\boldsymbol{x} ; \boldsymbol{\theta} + \boldsymbol{\epsilon}) = p(\boldsymbol{x} ; \boldsymbol{\theta}) &+ \sum_{j=1}^{d} \epsilon_j \frac{\partial p(\boldsymbol{x} ; \boldsymbol{\theta})}{\partial \theta_j} + \frac{1}{2!} \sum_{j,k=1}^{d} \epsilon_j \epsilon_k \frac{\partial^2 p(\boldsymbol{x} ; \boldsymbol{\theta})}{\partial \theta_j \partial \theta_k} \\
&+ \frac{1}{3!} \sum_{j,k,l=1}^{d} \epsilon_j \epsilon_k \epsilon_l \frac{\partial^3 p(\boldsymbol{x} ; \boldsymbol{\theta})}{\partial \theta_j \partial \theta_k \partial \theta_l} + \sum_{j,k,l,m=1}^{d} O(\epsilon_j \epsilon_k \epsilon_l \epsilon_m) ,
\end{aligned} \tag{82}$$

which can be rewritten in terms of the score function as

$$\begin{aligned}
\frac{p(\boldsymbol{x} ; \boldsymbol{\theta} + \boldsymbol{\epsilon})}{p(\boldsymbol{x} ; \boldsymbol{\theta})} = 1 &+ \sum_{i=1}^{d} \epsilon_j s_j(\boldsymbol{x} ; \boldsymbol{\theta}) + \frac{1}{2!} \sum_{j,k=1}^{d} \epsilon_j \epsilon_k U_{jk}(\boldsymbol{x} ; \boldsymbol{\theta}) \\
&+ \frac{1}{3!} \sum_{j,k,l=1}^{d} \epsilon_j \epsilon_k \epsilon_l V_{jkl}(\boldsymbol{x} ; \boldsymbol{\theta}) + \sum_{j,k,l,m=1}^{d} O(\epsilon_j \epsilon_k \epsilon_l \epsilon_m) ,
\end{aligned} \tag{83}$$

where

$$U_{jk}(\boldsymbol{x}\,;\boldsymbol{\theta}) \equiv \frac{1}{p(\boldsymbol{x}\,;\boldsymbol{\theta})}\,\frac{\partial^2 p(\boldsymbol{x}\,;\boldsymbol{\theta})}{\partial\theta_j\,\partial\theta_k}, \tag{84a}$$

$$V_{jkl}(\boldsymbol{x}\,;\boldsymbol{\theta}) \equiv \frac{1}{p(\boldsymbol{x}\,;\boldsymbol{\theta})}\,\frac{\partial^3 p(\boldsymbol{x}\,;\boldsymbol{\theta})}{\partial\theta_j\,\partial\theta_k\,\partial\theta_l}. \tag{84b}$$

Now let us consider the expectation of $\psi_{\boldsymbol{x},\boldsymbol{\theta},i}(\boldsymbol{\epsilon})$ under $\mathcal{P}$, conditional on $(\boldsymbol{x},\boldsymbol{\theta})$. If the kernel $K$ is i) sufficiently narrow and ii) vanishes sufficiently fast as $\boldsymbol{\epsilon}$ moves away from $\boldsymbol{0}$, then this expectation can be approximated (up to next-to-leading order in the kernel widths) using (83) and (81) as follows:

$$\mathrm{E}_{(\boldsymbol{x},\boldsymbol{\theta},\boldsymbol{\epsilon})\sim\mathcal{P}}\Big[\psi_{\boldsymbol{x},\boldsymbol{\theta},i}(\boldsymbol{\epsilon})\,\Big|\,\boldsymbol{x},\boldsymbol{\theta}\Big] \equiv \frac{\int d\boldsymbol{\epsilon}\,\mathcal{P}(\boldsymbol{\theta},\boldsymbol{\epsilon},\boldsymbol{x})\,\psi_{\boldsymbol{x},\boldsymbol{\theta},i}(\boldsymbol{\epsilon})}{\int d\boldsymbol{\epsilon}\,\mathcal{P}(\boldsymbol{\theta},\boldsymbol{\epsilon},\boldsymbol{x})} \tag{85a}$$

$$= \frac{\pi(\boldsymbol{\theta})\,p(\boldsymbol{x}\,;\boldsymbol{\theta})\displaystyle\int d\boldsymbol{\epsilon}\,K_{\boldsymbol{\theta}}(\boldsymbol{\epsilon})\,\frac{p(\boldsymbol{x}\,;\boldsymbol{\theta}+\boldsymbol{\epsilon})}{p(\boldsymbol{x}\,;\boldsymbol{\theta})}\,\psi_{\boldsymbol{x},\boldsymbol{\theta},i}(\boldsymbol{\epsilon})}{\pi(\boldsymbol{\theta})\,p(\boldsymbol{x}\,;\boldsymbol{\theta})\displaystyle\int d\boldsymbol{\epsilon}\,K_{\boldsymbol{\theta}}(\boldsymbol{\epsilon})\,\frac{p(\boldsymbol{x}\,;\boldsymbol{\theta}+\boldsymbol{\epsilon})}{p(\boldsymbol{x}\,;\boldsymbol{\theta})}} \tag{85b}$$

$$\approx \frac{\displaystyle\int d\boldsymbol{\epsilon}\left[K_{\boldsymbol{\theta}}(\boldsymbol{\epsilon})\,\psi_{\boldsymbol{x},\boldsymbol{\theta},i}(\boldsymbol{\epsilon})\left(\epsilon_i\,s_i(\boldsymbol{x}\,;\boldsymbol{\theta})+\frac{1}{6}\epsilon_i^3 V_{iii}(\boldsymbol{x}\,;\boldsymbol{\theta})+\frac{1}{2}\sum_{\substack{j=1\\j\neq i}}^{d}\epsilon_i\,\epsilon_j^2 V_{ijj}(\boldsymbol{x}\,;\boldsymbol{\theta})\right)\right]}{1+\displaystyle\int d\boldsymbol{\epsilon}\,\frac{K_{\boldsymbol{\theta}}(\boldsymbol{\epsilon})}{2}\sum_{j=1}^{d}\epsilon_j^2\,U_{jj}(\boldsymbol{x}\,;\boldsymbol{\theta})}. \tag{85c}$$

Up to leading order in the kernel widths, we have

$$\mathrm{E}_{(\boldsymbol{x},\boldsymbol{\theta},\boldsymbol{\epsilon})\sim\mathcal{P}}\Big[\psi_{\boldsymbol{x},\boldsymbol{\theta},i}(\boldsymbol{\epsilon})\,\Big|\,\boldsymbol{x},\boldsymbol{\theta}\Big] \approx s_i(\boldsymbol{x}\,;\boldsymbol{\theta})\int d\boldsymbol{\epsilon}\,K_{\boldsymbol{\theta}}(\boldsymbol{\epsilon})\,\psi_{\boldsymbol{x},\boldsymbol{\theta},i}(\boldsymbol{\epsilon})\,\epsilon_i \tag{86a}$$

$$= s_i(\boldsymbol{x}\,;\boldsymbol{\theta})\,\mathrm{E}_{\boldsymbol{\epsilon}\sim K_{\boldsymbol{\theta}}}\Big[\epsilon_i\,\psi_{\boldsymbol{x},\boldsymbol{\theta},i}(\boldsymbol{\epsilon})\Big], \tag{86b}$$

which leads to the kernel score approximation $\boldsymbol{s}^{\mathrm{KSA}}$ in (24).

# D   Narrowing Down the Choices for KSE

The quality of the kernel score estimate for $\boldsymbol{s}$ will depend on various choices including the kernel distribution $K_{\boldsymbol{\theta}}$, the kernel-width parameter $\boldsymbol{\lambda}(\boldsymbol{\theta})$, the difference function $\boldsymbol{\psi}_{\boldsymbol{x},\boldsymbol{\theta}}$, and the prior distribution $\pi(\boldsymbol{\theta})$. In this section we will provide optimal functional forms for $K_{\boldsymbol{\theta}}$ (up to the choice of the kernel-width) and $\boldsymbol{\psi}_{\boldsymbol{x},\boldsymbol{\theta}}$ from a bias–variance trade-off perspective.

## D.1   Bias of the Kernel Score Approximation

From (85c), we can show that, up to leading order in kernel widths, the error (bias) of $\boldsymbol{s}^{\mathrm{KSA}}$ can be written as

$$s_i^{\mathrm{KSA}}-s_i \approx \frac{V_{iii}}{6}\,\frac{\mathrm{E}_{\boldsymbol{\epsilon}\sim K_{\boldsymbol{\theta}}}\Big[\epsilon_i^3\,\psi_{\boldsymbol{x},\boldsymbol{\theta},i}(\boldsymbol{\epsilon})\Big]}{\mathrm{E}_{\boldsymbol{\epsilon}\sim K_{\boldsymbol{\theta}}}\Big[\epsilon_i\,\psi_{\boldsymbol{x},\boldsymbol{\theta},i}(\boldsymbol{\epsilon})\Big]}+\sum_{j\neq i}\frac{V_{ijj}}{2}\,\frac{\mathrm{E}_{\boldsymbol{\epsilon}\sim K_{\boldsymbol{\theta}}}\Big[\epsilon_i\,\epsilon_j^2\,\psi_{\boldsymbol{x},\boldsymbol{\theta},i}(\boldsymbol{\epsilon})\Big]}{\mathrm{E}_{\boldsymbol{\epsilon}\sim K_{\boldsymbol{\theta}}}\Big[\epsilon_i\,\psi_{\boldsymbol{x},\boldsymbol{\theta},i}(\boldsymbol{\epsilon})\Big]}-\frac{s_i}{2}\sum_{j=1}^{d}U_{jj}\,\mathrm{E}_{K_{\boldsymbol{\theta}}}\Big[\epsilon_j^2\Big]. \tag{87}$$

Here, $s_i$, $s_i^{\text{KSA}}$, $U_{ij}$, $V_{ijk}$ are all functions of $\boldsymbol{x}$ and $\boldsymbol{\theta}$.

## D.2   Local Variance of the Regression Target

From (24), we can show that, up to leading order in kernel widths, the variance of the regression target for a given input $(\boldsymbol{x}, \boldsymbol{\theta})$ is given by

$$\text{Var}_{(\boldsymbol{x},\boldsymbol{\theta},\boldsymbol{\epsilon})\sim\mathcal{P}}\left[\frac{\psi_{\boldsymbol{x},\boldsymbol{\theta},i}(\boldsymbol{\epsilon})}{\text{E}_{\boldsymbol{\epsilon}\sim K_{\boldsymbol{\theta}}}\left[\epsilon_i \, \psi_{\boldsymbol{x},\boldsymbol{\theta},i}(\boldsymbol{\epsilon})\right]}\;\middle|\; \boldsymbol{x},\boldsymbol{\theta}\right] = -\left[s_i^{\text{KSA}}(\boldsymbol{x};\boldsymbol{\theta})\right]^2 + \frac{\text{E}_{(\boldsymbol{x},\boldsymbol{\theta},\boldsymbol{\epsilon})\sim\mathcal{P}}\left[\psi_{\boldsymbol{x},\boldsymbol{\theta},i}^2(\boldsymbol{\epsilon})\;\middle|\;\boldsymbol{x},\boldsymbol{\theta}\right]}{\text{E}_{\boldsymbol{\epsilon}\sim K_{\boldsymbol{\theta}}}^2\left[\epsilon_i \, \psi_{\boldsymbol{x},\boldsymbol{\theta},i}(\boldsymbol{\epsilon})\right]}$$

(88a)

$$\approx -\left[s_i^{\text{KSA}}(\boldsymbol{x};\boldsymbol{\theta})\right]^2 + \frac{\text{E}_{\boldsymbol{\epsilon}\sim K_{\boldsymbol{\theta}}}\left[\psi_{\boldsymbol{x},\boldsymbol{\theta},i}^2(\boldsymbol{\epsilon})\right]}{\text{E}_{\boldsymbol{\epsilon}\sim K_{\boldsymbol{\theta}}}^2\left[\epsilon_i \, \psi_{\boldsymbol{x},\boldsymbol{\theta},i}(\boldsymbol{\epsilon})\right]}.$$

(88b)

The bias in (87) and the variance in (88b) both depend on the various choices mentioned above. In particular, wider kernels lead to larger bias and lower variance. Here we will optimize the choice of $K_{\boldsymbol{\theta}}$ and $\boldsymbol{\psi}_{\boldsymbol{x},\boldsymbol{\theta}}$ to get the best local variance for a given bias in $s^{\text{KSA}}$.

## D.3   Choosing the Kernel and Difference Function

We begin by imposing the following restrictions on $K_{\boldsymbol{\theta}}$ and $\boldsymbol{\psi}_{\boldsymbol{x},\boldsymbol{\theta}}$.

1. To avoid contributions from higher-order terms, **we will use kernels with bounded support**.

2. For simplicity, we will consider **product kernels** and **factorizable difference functions** of the following form:

$$K_{\boldsymbol{\theta}}(\boldsymbol{\epsilon}) = \prod_{i=1}^{d} \frac{1}{\lambda_i(\boldsymbol{\theta})} \, \kappa\left(\frac{\epsilon_i}{\lambda_i(\boldsymbol{\theta})}\right),$$

(89a)

$$\psi_{\boldsymbol{x},\boldsymbol{\theta},i}(\boldsymbol{\epsilon}) = \text{sign}(\epsilon_i) \, \alpha\left(\frac{|\epsilon_i|}{\lambda_i(\boldsymbol{\theta})}\right) \left[\prod_{j\neq i} \beta\left(\frac{|\epsilon_j|}{\lambda_j(\boldsymbol{\theta})}\right)\right],$$

(89b)

where $\kappa$ is a unit-normalized kernel with support in the range $[-1, 1]$.

Let $\mu_k$ be the $k$-th absolute moment of $\kappa$. Let us define the properties $a_{1,k}$, $a_{2,k}$, $b_{1,k}$, and $b_{2,k}$ as

$$\mu_k \equiv \int_{-1}^{1} dt \, \kappa(t) \, |t|^k,$$

(90a)

$$a_{1,k} \equiv \int_{-1}^{1} dt \, \kappa(t) \, \alpha(|t|) \, |t|^k, \qquad a_{2,k} \equiv \int_{-1}^{1} dt \, \kappa(t) \, \alpha^2(|t|) \, |t|^k,$$

(90b)

$$b_{1,k} \equiv \int_{-1}^{1} dt \, \kappa(t) \, \beta(|t|) \, |t|^k, \qquad b_{2,k} \equiv \int_{-1}^{1} dt \, \kappa(t) \, \beta^2(|t|) \, |t|^k.$$

(90c)

With the choices for $K_{\boldsymbol{\theta}}$ and $\psi_{\boldsymbol{x},\boldsymbol{\theta}}$ in (89) and the definitions in (90), the bias and local variance can be written as

$$s_i^{\text{KSA}} - s_i \approx \frac{V_{iii}}{6} \lambda_i^2(\boldsymbol{\theta}) \frac{a_{1,3}}{a_{1,1}} + \sum_{j\neq i} \frac{V_{ijj}}{2} \lambda_j^2(\boldsymbol{\theta}) \frac{b_{1,2}}{b_{1,0}} - \frac{s_i}{2} \sum_{j=1}^{d} U_{jj} \lambda_j^2(\boldsymbol{\theta}) \mu_2, \quad (91)$$

$$\text{Var}_{(\boldsymbol{x},\boldsymbol{\theta},\boldsymbol{\epsilon})\sim\mathcal{P}} \left[ \frac{\psi_{\boldsymbol{x},\boldsymbol{\theta},i}(\boldsymbol{\epsilon})}{\text{E}_{\boldsymbol{\epsilon}\sim K_{\boldsymbol{\theta}}}\left[\epsilon_i \, \psi_{\boldsymbol{x},\boldsymbol{\theta},i}(\boldsymbol{\epsilon})\right]} \,\middle|\, \boldsymbol{x},\boldsymbol{\theta} \right] + \left[s_i^{\text{KSA}}(\boldsymbol{x};\boldsymbol{\theta})\right]^2 \approx \frac{1}{\lambda_i^2(\boldsymbol{\theta})} \frac{a_{2,0}}{a_{1,1}^2} \left(\frac{b_{2,0}}{b_{1,0}^2}\right)^{d-1}. \quad (92)$$

We want to minimize the bias for a given variance, say $C_i$:

$$C_i = \frac{1}{\lambda_i^2(\boldsymbol{\theta})} \frac{a_{2,0}}{a_{1,1}^2} \left(\frac{b_{2,0}}{b_{1,0}^2}\right)^{d-1}. \quad (93)$$

Now the bias can be written as

$$s_i^{\text{KSA}} - s_i \approx \frac{a_{2,0}}{a_{1,1}^2} \left(\frac{b_{2,0}}{b_{1,0}^2}\right)^{d-1} \left[ \frac{V_{iii}}{6\,C_i(\boldsymbol{\theta})} \frac{a_{1,3}}{a_{1,1}} + \sum_{j\neq i} \frac{V_{ijj}}{2\,C_j(\boldsymbol{\theta})} \frac{b_{1,2}}{b_{1,0}} - \sum_{j=1}^{d} \frac{s_i\,U_{jj}}{2\,C_j(\boldsymbol{\theta})} \mu_2 \right]. \quad (94)$$

There are three terms in this bias. Since

$$\int d\boldsymbol{x}\, p(\boldsymbol{x};\boldsymbol{\theta})\, V_{ijj}(\boldsymbol{x};\boldsymbol{\theta}) = 0, \qquad \forall i,j, \quad (95)$$

the bias contributions from the first two terms in (94) will be suppressed when averaging over a large observed dataset produced at $\boldsymbol{\theta}_{\text{true}}$ close to $\boldsymbol{\theta}$. For this reason, we will prioritize the minimization of the magnitude of the summand in the third term, namely

$$\left(\frac{b_{2,0}}{b_{1,0}^2}\right)^{d-1} \frac{a_{2,0}}{a_{1,1}^2} \mu_2 \, \frac{s_i\,U_{jj}}{2\,C_j(\boldsymbol{\theta})}.$$

Here $C_j$ is fixed, and $s_i$ and $U_{jj}$ are *a priori* unknown properties of the distribution $p(\boldsymbol{x};\boldsymbol{\theta})$. For any choice of the kernel $\kappa$ and function $\alpha$,

$$\frac{b_{2,0}}{b_{1,0}^2} \geq 1, \quad (96)$$

with equality if $\beta$ is a constant function.[12] This suggests choosing $\beta$ to be a constant function. Now consider

$$a_{2,0}\,\mu_2 - a_{1,1}^2 = \int_{-1}^{1} dt\, \kappa(t) \int_{-1}^{1} dt'\, \kappa(t') \left[\alpha^2(|t|)\,|t'|^2 - \alpha(|t|)\,|t|\,\alpha(|t'|)\,|t'|\right] \quad (97a)$$

$$= \frac{1}{2} \int_{-1}^{1} dt\, \kappa(t) \int_{-1}^{1} dt'\, \kappa(t') \left[\alpha^2(|t'|)\,|t|^2 + \alpha^2(|t|)\,|t'|^2 - 2\,\alpha(|t|)\,|t|\,\alpha(|t'|)\,|t'|\right] \quad (97b)$$

$$= \frac{1}{2} \int_{-1}^{1} dt \int_{-1}^{1} dt' \left[\alpha(|t'|)\,|t| - \alpha(|t|)\,|t'|\right]^2 \geq 0, \quad (97c)$$

---

[12]Equality will also hold if $|t|$ is a constant under $t \sim \kappa$. In this case, $\beta(|t|)$ will trivially be a constant function.

$$\Rightarrow \quad \frac{a_{2,0}}{a_{1,1}^2} \mu_2 \geq 1, \tag{97d}$$

with equality if $\alpha(|t|) \propto |t|$.[13] This suggests setting $\alpha$ to be proportional to $|t|$. Using these choices for $\alpha$ and $\beta$, we get the following simple form of $\psi_{x,\theta,i}$

$$\psi_{x,\theta,i}(\epsilon) \equiv \psi_i^{\text{identity}}(\epsilon) = \epsilon_i. \tag{98}$$

Note that scaling the choice of $\psi_{x,\theta,i}(\epsilon)$ by a multiplicative factor independent of $\epsilon$ (but possibly dependent on $x$, $\theta$, or $i$) will not tangibly affect the KSE procedure. Under this linear choice for $\psi$, the bias and local variance terms become

$$s_i^{\text{KSA}} - s_i \approx \frac{V_{iii}}{6} \lambda_i^2(\theta) \frac{\mu_4}{\mu_2} + \sum_{j \neq i} \frac{V_{ijj}}{2} \lambda_j^2(\theta) \mu_2 - \sum_{j=1}^{d} \frac{s_i U_{jj}}{2} \lambda_j^2(\theta) \mu_2, \tag{99}$$

$$\text{Var}_{(x,\theta,\epsilon)\sim\mathcal{P}} \left[ \frac{\psi_{x,\theta,i}(\epsilon)}{\text{E}_{\epsilon\sim K_\theta} \left[ \epsilon_i \, \psi_{x,\theta,i}(\epsilon) \right]} \,\middle|\, x,\theta \right] \approx \frac{1}{\lambda_i^2(\theta) \mu_2} - \left[ s_i^{\text{KSA}}(x \,;\, \theta) \right]^2. \tag{100}$$

Next we repeat the process of minimizing the bias for a given local variance (focusing, this time, on the first and second terms of the bias) to identify a good choice of $\kappa$. Fixing $\lambda_i^2(\theta)\mu_2$ for all $i$ only leaves the first term of the bias undetermined. This can be minimized by choosing a $\kappa$ that minimizes $\mu_4/\mu_2^2$. We have

$$\frac{\mu_4}{\mu_2^2} - 1 = \frac{\text{E}_{t\sim\kappa}\left[|t|^4\right] - \text{E}_{t\sim\kappa}^2\left[|t|^2\right]}{\mu_2^2} = \frac{\text{Var}_{t\sim\kappa}\left[|t|^2\right]}{\mu_2^2} \geq 0, \tag{101}$$

with equality only if $|t|$ is a constant under $\kappa$, i.e., if $\kappa$ is a symmetric delta function kernel. In summary, in this appendix, we have argued in favor of using the delta kernel in eq. (19) and the linear difference function in (22) for KSE based on bias–variance trade-off considerations.

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
