# Peer review of "New Machine Learning Techniques for Simulation-Based Inference: InferoStatic Nets, Kernel Score Estimation, and Kernel Likelihood Ratio Estimation"

_SciPost Physics, doi:SciPost Phys. Codebases 14 (2023) , SciPost Phys. Codebases 14-r0.1 (2023)_

## Round 1 · Referee Report · Ramon Winterhalder (Referee 1) · 2022-11-8

Strengths

The usage of a "backend" function phi, which obeys several useful properties as shown in eq (10) instead of learning the score and the ratio directly, shows great potential. Furthermore, baking-in symmetries into the network architecture that are fulfilled precisely and do not have to be learned are a great idea, as it generally increases the precision and, most importantly, the stability of the network.

Weaknesses

Missing explicit LHC examples.

Report

Thanks to the author for the interesting paper. The presented approaches and the results are exciting and offer great potential for future applications for LHC analyses. For this reason, the article is worth publishing. However, as the paper does not explicitly showcase any physics example, I would either ask the others to consider some LHC examples and add them to their draft or recommend the paper for the partner journal SciPost Physics Codebases. Further, I would ask the authors to add another citation (see the requested changes).

Requested changes

Before publishing, I would ask the others to cite another method of directly estimating the likelihood using the Matrix-Element Method (2210.00019), which would fit into Table 1 next to the NDE approach.

  • validity: top
  • significance: high
  • originality: high
  • clarity: high
  • formatting: excellent
  • grammar: excellent

Author:  Prasanth Shyamsundar  on 2023-02-14  [id 3353]

(in reply to Report 1 by Ramon Winterhalder on 2022-11-08)

Hey Ramon,

Thanks a lot for your feedback and suggestions! We have added the citation as suggested.

Several papers in the last few years have already showcased the applications of score and likelihood ratio estimation in particle physics. We believe that redoing physics examples will only lead to duplication of effort, without significantly improving the value of the manuscript, which is already very lengthy as it is. We'll leave the decision of the appropriate journal to the editor.

Regards,
Prasanth (on behalf of the authors)

---

## Round 2 · List of Changes

We have added a citation as suggested by the referee. We have also fixed a few typos and made minor updates to the text to improve readability.

---

## Editorial Decision

published